# NOBLE – Neural Operator with Biologically-informed Latent Embeddings to Capture Experimental Variability in Biological Neuron Models

**Luca Ghafourpour**[1,2]     **Valentin Duruisseaux**[2*]     **Bahareh Tolooshams**[3,4*]     **Philip H. Wong**[5]
**Costas A. Anastassiou**[5,6]     **Anima Anandkumar**[2]

[1]ETH Zürich     [2]California Institute of Technology     [3]University of Alberta
[4]Alberta Machine Intelligence Institute (Amii)     [5]Cedars-Sinai Medical Center
[6]Archimedes AI, Athena Research Center

## Abstract

Characterizing the cellular properties of neurons is fundamental to understanding their function in the brain. In this quest, the generation of bio-realistic models is central towards integrating multimodal cellular data sets and establishing causal relationships. However, current modeling approaches remain constrained by the limited availability and intrinsic variability of experimental neuronal data. The deterministic formalism of bio-realistic models currently precludes accounting for the natural variability observed experimentally. While deep learning is becoming increasingly relevant in this space, it fails to capture the full biophysical complexity of neurons, their nonlinear voltage dynamics, and variability. To address these shortcomings, we introduce NOBLE, a neural operator framework that learns a mapping from a continuous frequency-modulated embedding of interpretable neuron features to the somatic voltage response induced by current injection. Trained on synthetic data generated from bio-realistic neuron models, NOBLE predicts distributions of neural dynamics accounting for the intrinsic experimental variability. Unlike conventional bio-realistic neuron models, interpolating within the embedding space offers models whose dynamics are consistent with experimentally observed responses. NOBLE enables the efficient generation of synthetic neurons that closely resemble experimental data and exhibit trial-to-trial variability, offering a $4200\times$ speedup over the numerical solver. NOBLE is the first scaled-up deep learning framework that validates its generalization with real experimental data. To this end, NOBLE captures fundamental neural properties in a unique and emergent manner that opens the door to a better understanding of cellular composition and computations, neuromorphic architectures, large-scale brain circuits, and general neuroAI applications.

## 1 Introduction

Hundreds of distinct neuronal cell types co-exist and compute within neural circuits, yet how they shape cognitive functions remains essentially unanswered [1–5]. This is particularly true in the human brain, where access and monitoring capabilities are severely limited compared to animal models. Over the past decade, multimodal cellular datasets that integrate electrophysiology, morphology, and transcriptomics have emerged for human cell types [6–10].

---

*These authors contributed equally to this work.

39th Conference on Neural Information Processing Systems (NeurIPS 2025).

While integrating across the different modalities remains a challenge, clear differences in gene expression, morphology, and electrophysiology are evident across cell types. However, understanding how these differences impact brain processing is crucial, e.g. to uncover how expression of specific genes relates to neurological diseases.

Cellular models representing multiple data modalities are invaluable as they offer a degree of control and perturbations that are experimentally impossible (e.g., [11–13]). Recently, evolutionary multi-objective optimization algorithms [14] have been used to generate and validate bio-realistic models of neurons in the form of 3D multi-compartment partial differential equation (PDE) models that mirror both their shape and ion channel expression, shaping their electrical properties [12, 13, 15, 16]. Yet, such models are deterministic and fail to capture the intrinsic variability observed experimentally, where identical input to the same neuron often results in different electrophysiological responses. One approach is to generate families of models, referred to as "hall-of-fame" (HoF) models [12, 13, 17] to represent a single cell. While each HoF model is distinct and reproduces the electrophysiological features of parts of an experiment, the ensemble of deterministic HoF models is used as a collective representation that captures both the main features as well as their variability in an experiment [12, 13]. Typically, neurons exhibit highly nonlinear behavior, necessitating equally complex models rendering the optimization computationally demanding (i.e. requiring about 600k CPU core hours per single-neuron model [12, 13]). Yet, even tiny perturbations of the model parameters lead to large deviations from experimental data [18]. Other approaches have explored capturing variability through introducing stochasticity in neuron models [19–21]. However, the synthetic injection of white noise is non-mechanistic and introduces perturbations that can lead to unrealistic predictions [22–24]. In summary, capturing the nature and variability of neurons is a challenge with existing computational techniques.

The challenges of scalability and the computational cost associated with traditional numerical modeling approaches, such as numerical integrators and evolutionary optimization algorithms, have led the scientific community to explore the use of machine learning to accelerate simulations by learning underlying relationships between variables directly from experimental and synthetic data. While neural networks have been used successfully for many applications, they learn mappings between finite-dimensional vectors, which can limit their ability to model physical phenomena that are better described using functions in infinite-dimensional spaces and functional relationships between them [25]. As a result, neural networks can overfit to the training discretization and suffer from limited out-of-distribution capabilities. Neural operators [25–27] are a principled way to generalize neural networks to learn operators mapping functions to functions, with a universal operator approximation property [28]. A variety of neural operators have been proposed, such as the Fourier Neural Operator (FNO) [29, 30].

Machine learning approaches have been applied to model single-cell electrophysiology, primarily for point-neuron systems such as FitzHugh–Nagumo [31, 32] and Hodgkin–Huxley [33]. Conventional neural networks [34–36] and physics-informed neural networks [37–42] successfully reproduced their dynamics but remain highly specific to deterministic formulations and require retraining for each new stimulus. More recently, neural operators demonstrated strong potential for learning the governing dynamics of Hodgkin–Huxley systems [43], though the study was limited to simplified data, without capturing biological variability. Related works like NeuPRINT [44] captured biological neuronal variability, but on slower in vivo 2-photon calcium imaging data and models population-level fluorescence dynamics, rather than fast intracellular voltage dynamics of individual neurons.

We build on these advances and address current limitations to enable deeper insights into brain function and neuroAI.

**Contributions.** We introduce NOBLE (Neural Operator with Biologically-informed Latent Embeddings), a neural operator framework for learning the nonlinear somatic dynamics across a population of HoF models for a single neuron (Figure 1). NOBLE is the first scaled-up deep learning framework whose performance is validated with experimental human cortex data. Rather than training a separate independent surrogate for each HoF model, NOBLE learns a single neural operator that maps from a continuous latent space of user-defined, interpretable neuron characteristics to an ensemble of somatic voltage responses induced by current injection. This latent space is constructed using an embedding strategy informed by the specified characteristics of the neuron models.

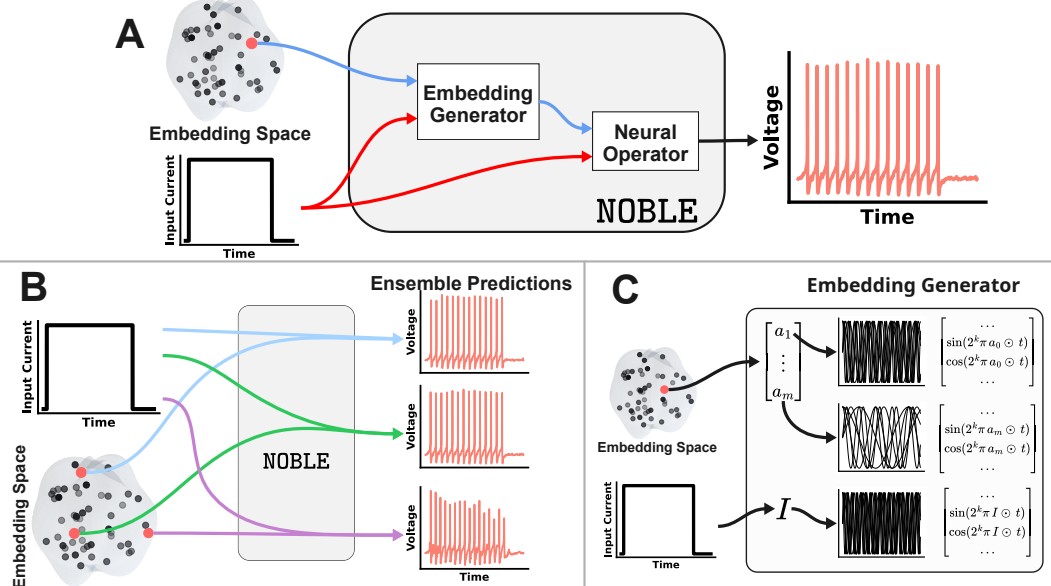

Figure 1: The Neural Operator with Biologically-informed Latent Embeddings (NOBLE) framework.
**A**) In NOBLE, a current injection and neuron model features are first encoded using the proposed embedding strategy, before passing through a neural operator to produce a prediction for the somatic voltage response. **B**) NOBLE can be queried in parallel with different model latent representations to produce ensemble predictions. **C**) The proposed embedding in NOBLE encodes specified neuron features and the input current as a stack of trigonometric time-series, as described in Section 3.4.

As an example application, we train and evaluate NOBLE on a parvalbumin-positive (PVALB) neuron dataset generated using 50 HoF PVALB models (Figure 2). We show that a single NOBLE model accurately captures both subthreshold and spiking dynamics across all 50 HoF models (in-distribution) as well as 10 unseen HoF models (out-of-distribution) while achieving a significant speedup of 4200× over the numerical solver used to generate the dataset (Figure 3). In addition, the NOBLE predictions across 16 electrophysiological features of interest (including spike count, amplitude, and width) remain within the variability observed in experimental data (Figure 5B). Additional ablation studies confirm that biologically informed embeddings are critical for capturing both firing and non-firing dynamics, and demonstrate that we can enhance performance on targeted features without compromising overall dynamics by introducing a feature-specific fine-tuning approach.

We further show that NOBLE can successfully generate novel bio-realistic neuron models by sampling and interpolating within the latent space of models. The dynamics of novel neuron models generated by NOBLE align both with previously unseen HoF PVALB models and experimentally observed somatic responses. In contrast, direct interpolation between the parameters of bio-realistic PDE-based neuron models fails due to the sensitivity and nonlinearity of the underlying PDEs [45, 46] (Figure 4). We also successfully instantiate and train an additional NOBLE on a vasoactive intestinal peptide (VIP) interneuron to demonstrate the generalizability of our embedding framework. Owing to NOBLE's ability to generate novel bio-realistic neuron models, ensemble predictions are no longer constrained to the original 50 HoF models used for training. We demonstrate that NOBLE can produce somatic voltage responses for an arbitrary number of biologically plausible neurons by predicting responses to input stimuli across a larger set of models (Figure 5). The results showcase NOBLE's ability to accurately capture a broad range of neuron dynamics while enabling dense interpolation across model space. By unlocking the efficient, unlimited generation of diverse yet realistic neurons from a continuous embedding, NOBLE offers a scalable alternative to computationally intensive, scale-limited evolutionary approaches, laying the foundation for brain-scale neural circuit modeling.

Finally, the biologically-informed latent representation of the neuron models together with the capability of NOBLE to generate arbitrarily many new bio-realistic neuron models also offers further insight into the behavior of neural dynamics. We can use NOBLE to obtain the somatic responses to current injections on a fine grid in the model latent representation space, and consequently construct heat maps and surface plots to better understand how neuron features used for the model latent representation affect any electrophysiological feature of interest.

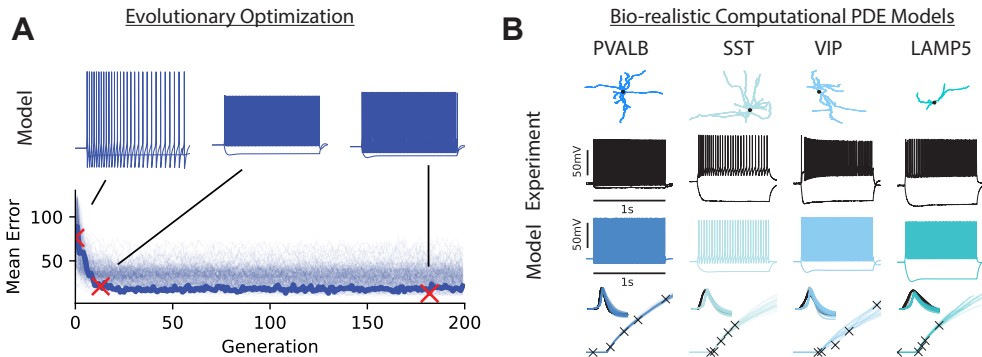

Figure 2: Creation of Bio-realistic PDE-based Neuron Models. **A**) Evolutionary optimization process for a neuron of interest, with voltage responses sampled at different generations (top) and the error history with other experimental neurons overlaid in the background (bottom). **B**) Sample HoF models of various inhibitory cell-types, showing morphology (top), experimental voltage traces (2nd row), simulated voltage traces (3rd row), and spike waveform and frequency-current curves (bottom).

## 2 Background on Bio-Realistic Neuron Modeling

We create bio-realistic PDE-based neuron models (based on the cable equation [47]) using actual reconstructions of neuron morphologies from human cortical data [6, 8, 9] (Figure 2). We instantiate these models using a framework [45] built on the NEURON simulation environment [46], which uses a spatial discretization to simulate the models as a system of coupled ordinary differential equations. We place ion channels in an "all-active" configuration [12, 13], where active ion channels are distributed along both somatic and dendritic compartments along the neuron morphology. For each experimental neuron, models are generated using a multi-objective evolutionary optimization framework [13] to find ion conductance parameters replicating a standard set of electrophysiological features from patch clamp recordings (Figure 2A). We adopt a two-stage optimization strategy, first fitting passive subthreshold responses, followed by capturing the active dynamics above the spiking threshold and the full frequency-current curve of each neuron. After 250 generations of evolutionary optimization, the models that best minimize the mean z-score error between simulated and actual experimental electrophysiological features are selected as HoF models (Figure 2B). More details about the electrophysiological features of interest are provided in Appendix B.

To illustrate the proposed approach, we consider a randomly selected PVALB human cortical neuron, for which we created 60 HoF models. PVALB neurons are fast-spiking inhibitory interneurons regulating high-frequency gamma oscillations (30-80Hz) [48] and their dysfunction has been associated with cognitive impairments such as schizophrenia and Alzheimer's disease [49, 50]. We also consider the class VIP of inhibitory interneurons, known for its disinhibitory role in cortical circuits [51–53].

## 3 Method

### 3.1 Subsampling

NOBLE utilizes the notable property of neural operators of training on low-resolution data while reserving the capability to generate dynamics at higher resolution. In this regard, we subsample the reference HoF simulations in time. To avoid discarding high-resolution information necessary for capturing neuron features of interest, we analyze how these features are affected by different subsampling factors and strategies, in particular via the discrepancy between HoF simulations and experimental data. We consider (i) low-pass filtering followed by decimation in time, (ii) low-pass filtering followed by truncation in the frequency domain, (iii) truncation in the frequency domain, and (iv) decimation in time without filtering. Across neuron features, we observe no consistent differences in performance between these strategies and thus opt for low-pass filtering followed by decimation in time. Our analysis (see Appendix C.1) reveals that 3× subsampling preserves the fidelity of extracted features, within the bounds of the intrinsic discrepancy between HoF simulations and experimental data, and without inducing notable aliasing. For the HoF simulations, we consider time series of $515\text{ms}$ with a timestep of $0.02\text{ms}$. For such signals, the subsampling reduces the sequence length from 25,750 to 8,583, substantially decreasing the computational load without compromising biological realism.

## 3.2 Current Amplitude Sampling

We consider square DC step current inputs, which are widely used in electrophysiological experiments and common for characterizing neuron behavior. We sample the current amplitudes from a skew-normal distribution whose support matches the experimentally validated range of the HoF models, $I \in [-0.11, 0.28]$nA. To effectively capture the highly nonlinear dynamics around the spiking threshold (0 to $0.05$nA) where neural responses transition abruptly from being non-spiking to spiking, the mode of our sampling distribution is strategically located within this peri-threshold window. To address the greater learning challenge posed by the high-frequency components of depolarizing, spiking responses (characterized by features such as spike width, latency to first spike, and spike count), we deliberately use a heavier positive tail in our sampling distribution. This ensures the model encounters numerous examples of spike onset and complex spiking patterns during training while still covering the full input range. The distribution of square-pulse amplitudes is shown in Figure 8.

## 3.3 Neural Operators for Neuron Dynamics Simulation

We choose to use neural operators as they offer clear advantages for modeling complex dynamics (Appendix C.3). Among neural operators, the Fourier Neural Operator (FNO) [29, 30] is very efficient as it leverages fast Fourier transforms on equidistant grids, which aligns naturally with our setting where both experimental recordings and PDE simulations are sampled at constant timesteps. FNOs provide a principled and efficient framework for modeling neuronal dynamics by learning mappings from input currents to voltage responses across a broad family of neuron models and current injections. Unlike conventional neural networks that operate on vector inputs and outputs of fixed sizes, the FNO learns operators, that is, mappings between functions. By operating in the frequency domain, the FNO efficiently captures global, nonlinear, and high-frequency components of voltage responses. These properties allow the model to generalize across different temporal resolutions, input currents, and neuron types, enabling the accurate simulation of unseen configurations without retraining.

## 3.4 Embedding Strategy for Neuron-Model Variability

NOBLE learns a single neural operator that maps from a continuous latent space of user-defined, interpretable neuron characteristics to an ensemble of somatic voltage responses induced by current injection. The frequency-current (F-I) curve is a useful electrophysiological descriptor that summarizes cellular excitability by relating injected current amplitude to the neuron's firing rate [54]. Differences between HoF parameterizations manifest as shifts in key features of this curve: the threshold current $I_{thr}$ (the minimum amplitude that elicits spiking) and the local slope $s_{thr}$ at $I_{thr}$ (the rate of increase in the firing rate upon spiking). Figure 3A displays examples of F-I curves for different HoF PVALB models, illustrating how variability in $I_{thr}$ and $s_{thr}$ can represent the trial-to-trial intrinsic variability observed when a single neuron is repeatedly recorded under the same current injection.

Using this observation, we propose representing a given neuron model by its threshold current $I_{thr}$ and local slope $s_{thr}$, that is, using $(I_{thr}, s_{thr})$. We propose to use this representation as part of a NeRF-style (Neural Radiance Field) embedding [55], where input features are encoded using sine and cosine functions. More precisely, a feature $p$ is encoded as a stack of trigonometric time-series

$$\gamma(p, t) = \left[ \sin(2^0 \pi p \odot t), \cos(2^0 \pi p \odot t), \ldots, \sin(2^{K-1} \pi p \odot t), \cos(2^{K-1} \pi p \odot t) \right], \quad (1)$$

for some integer $K > 0$, where the frequencies are modulated by the feature $p$. Here $t$ denotes the discretized time coordinates and $\odot$ indicates element-wise multiplication with appropriate broadcasting.

The use of sine and cosine functions for encoding features is particularly synergistic with FNOs, which operate in the frequency domain to learn mappings between functions. FNOs leverage the Fourier transform to represent and manipulate data as sums of sine and cosine functions, effectively learning complex patterns by capturing interactions among frequency components. NeRF-style encodings lead to a representation of the input features that aligns naturally with the spectral approach of FNOs, enhancing their ability to learn high-frequency dynamics. In this context, the sinusoidal embeddings can be thought of as a form of spectral lifting, translating low-dimensional inputs into a richer representation in the frequency domain that FNOs can more efficiently process.

We encode separately the model features $I_{thr}$ and $s_{thr}$, and the amplitude of the current injection, and stack the resulting embeddings as input channels. To compress the large range of $I_{thr}$ and $s_{thr}$ values into a manageable scale for embedding, we normalize them to $[0.5, 3.5]^2$, supporting more distinct feature space representations of HoF models. Figure 10 displays the latent representations in normalized $(I_{thr}, s_{thr})$-space of the 60 HoF PVALB models used in our numerical experiments.

### 3.5 `NOBLE`: Neural Operator with Biologically-informed Latent Embeddings

We introduce the Neural Operator with Biologically-informed Latent Embeddings (`NOBLE`), for modeling neuronal voltage dynamics in response to current injections. `NOBLE` offers a scalable alternative to computationally intensive numerical solvers for biophysically detailed, PDE-based neuron models. It learns a direct mapping from input currents and a continuous, interpretable latent space of neuron features, to the resulting voltage traces (Figure 1A). A key feature of `NOBLE` is its use of biologically-informed embeddings, which enables interpretability and generalization across biological neuron models. At its core, `NOBLE` is based on a neural operator, whose discretization invariance allows `NOBLE` to learn on low-resolution data and infer somatic voltage dynamics at higher resolutions. By combining a neural operator with the proposed continuous interpretable embedding, `NOBLE` learns a continuous operator over the space of bio-realistic neuron models.

The proposed `NOBLE` framework offers key advantages that set it apart from existing approaches:

- `NOBLE` provides a unified framework that learns ensemble dynamics directly, enabling it to generate diverse, biophysically plausible membrane potentials for the same input. Conditioned on a particular electrophysiological feature, it produces one realization of the intrinsic variability observed in biological neurons. This stands in contrast to previous deep learning approaches, which are inherently deterministic and produce a single trace for each input, failing to capture the trial-to-trial variability observed experimentally. To account for different neuronal behaviors, such models must be retrained for each variation, resulting in inefficiency and fragmentation.

- Through the latent embedding space of electrophysiological features, `NOBLE` can interpolate between known HoF models to produce new, bio-realistic neuronal responses. This capability is significant because HoF models are restricted to the finite set discovered by evolutionary optimization, and direct interpolation between their parameters does not result in realistic traces. As shown in Figure 4, interpolation in `NOBLE` 's latent space consistently produces valid, bio-realistic responses, whereas interpolations between PDE parameters do not.

- `NOBLE` can rapidly generate arbitrarily many distinct neuron models by sampling points within this continuous latent embedding space and producing the corresponding dynamics. This enables a single model to capture both spiking and subthreshold behaviors beyond the finite set of HoF models, providing an effectively infinite ensemble of bio-realistic responses that remain consistent with the variability observed in biological neurons. This is distinct from previous deep-learning methods, which were limited to predicting either spiking or subthreshold regimes in isolation.

- The bio-informed latent space of `NOBLE`, combined with its ability to generate unlimited realistic neuron models, enables fine-grained exploration of neural dynamics. By sampling models across this space, `NOBLE` can produce somatic responses for different latent features, and reveal how they influence electrophysiological behavior via visualizations (e.g. heat maps and surface plots).

## 4 Results

### 4.1 Experimental Setup

For evaluating `NOBLE`, we focus on the PVALB neuron example introduced in Section 2, and further assess the framework's generality using a VIP neuron. In the PVALB setting, `NOBLE` receives as input the applied current injection $I$, together with stacked embeddings of $I$ (with $K = 9$ different frequencies) and of the normalized model features $(I_{thr}, s_{thr})$ associated to a neuron model $\text{HoF}_\ell$ (with $K = 1$ frequency). `NOBLE` then outputs a corresponding somatic voltage response. The current injections are square DC steps with an activation duration of $400$ms, consistent across all stimuli used for training and testing. We have access to 60 HoF models, where 50 are used during training, $\{\text{HoF}^{train}\}$, and the remaining 10 HoF models $\{\text{HoF}^{test}\}$ are used for testing. Figure 10 displays the latent representations in normalized $(I_{thr}, s_{thr})$-space of these HoF models. For more details on the data generation, see Appendix D.1.

We use a 1D FNO (implemented as in the NeuralOperator library [56]) with 12 layers, each with 24 hidden channels and 256 Fourier modes. The resulting `NOBLE` with 1.8M trainable parameters is trained in PyTorch for 300 epochs using the Adam optimizer with learning rate $0.004$, and the ReduceLROnPlateau scheduler with factor $0.4$ and patience 4. The training minimizes the relative L4 error, while the performance metrics are reported using the relative L2 error for interpretability.

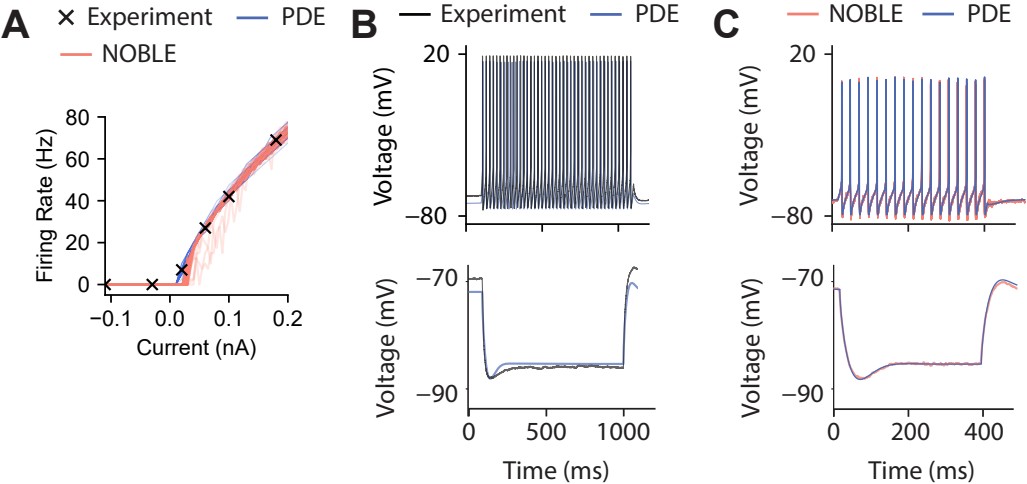

Figure 3: **A**) F–I curves from experimental recordings, PDE simulations, and `NOBLE` predictions on $\{\text{HoF}^{train}\}$. For one HoF model, **B**) compares experimental voltage responses with PDE simulations at a current injection of $0.1$nA (top) and $-0.11$nA (bottom), and **C**) compares the corresponding PDE simulations with the `NOBLE` predictions for the same HoF model.

To better evaluate physiologically meaningful behavior, we also report errors on four key electrophysiological features: `spikecount`, `AP1_width`, `mean_AP_amplitude`, and `steady_state_voltage` (see Appendix B for definitions of the features). For benchmarking, we compare `NOBLE` predictions against numerical simulations obtained from HoF models and experimental data since the HoF models were optimized to produce the closest approximations to experimental recordings and capture the biological variability required for a meaningful benchmark. Further details on the evaluation and evaluation metrics are provided in Appendix D.2.

The PyTorch codes used for our implementation of `NOBLE` and the numerical experiments are based on the NeuralOperator library [56], and are made available at [github.com/neuraloperator/noble](github.com/neuraloperator/noble).

## 4.2 Testing on HoF Models Included in the Training Set

We first validate that the trained `NOBLE` can accurately reproduce the somatic voltage responses of the training $\{\text{HoF}^{train}\}$ models when tested on current injections not seen during training. Figure 3C shows that the voltage traces exhibit minimal differences, confirming that `NOBLE` generalizes well to unseen inputs. This is supported by a relative L2 test error of $2.18\%$ with the $\{\text{HoF}^{train}\}$ models. Figure 3B also shows that the numerical solver outputs align closely with experimental recordings, and together with Figure 3C, indicates that `NOBLE` inherits this agreement and captures physiologically meaningful dynamics. Note that the available experimental recordings correspond to stimuli with activation durations of 1s. `NOBLE` also achieves errors of $3\%$ for `spikecount`, $32.8\%$ for `AP1_width`, $8.54\%$ for `mean_AP_amplitude`, and $0.83\%$ for `steady_state_voltage`. To further assess `NOBLE`'s applicability across neuron types, we trained it on a VIP neuron using the same architecture and observed similarly strong performance (see Appendix D.3 for more details).

The relatively higher error on `AP1_width` arises from how the feature is computed: it measures the width of the first spike at half amplitude, where the half level is defined between the spike peak and the subsequent after-hyperpolarization minimum. If the predicted peak or minimum is slightly misaligned relative to the ground truth, the half-voltage reference shifts, and the measured width corresponds to a different portion of the trace. This sensitivity makes `AP1_width` less stable to small deviations, so its relative error should be interpreted with caution compared to the other features.

We also generate the F-I curves using the trained `NOBLE` for $\{\text{HoF}^{train}\}$ and compare them with the reference F-I curves produced by the numerical solver for the same HoF models. As shown in Figure 3A, the curves from both methods remain close overall, although for 3 out of the 50 HoF models the `NOBLE` predictions show larger deviations in firing rate between $0.0 - 0.1$nA.

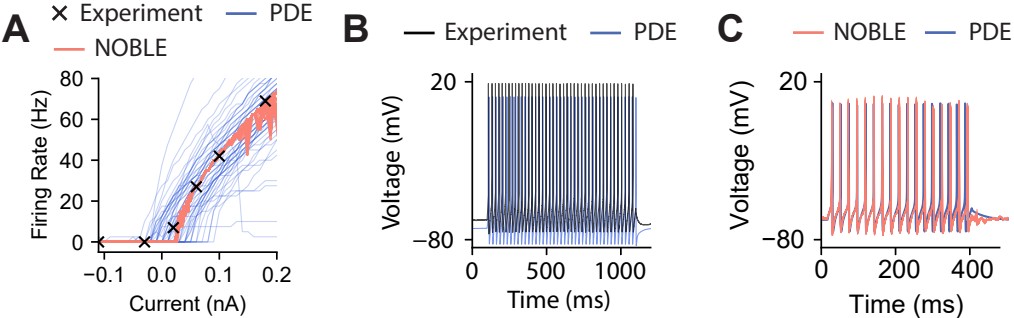

Figure 4: **A**) F-I curves from experimental recordings, PDE simulations, and NOBLE predictions on 50 interpolated HoF models. **B**) Experimental response vs. numerical simulation after interpolating in between PDE parameterizations. **C**) Numerical simulation of $\text{HoF}_k^{test}$ vs. distribution of 50 NOBLE predictions after interpolating within the model latent space near the $(I_{thr}, s_{thr})$ features of $\text{HoF}_k^{test}$.

### 4.3 Interpolating Between Models

We now test NOBLE's ability to interpolate between HoF models within the convex hull $\mathcal{CH}_{train}$ of the $\{\text{HoF}^{train}\}$ models used for training. Consider a PDE model $\text{HoF}_k^{test}$ excluded during training.

We first randomly sample 50 unseen synthetic models from a local neighborhood in the model latent space near the defining $(I_{thr}, s_{thr})$ features of $\text{HoF}_k^{test}$ (Figure 11 illustrates this neighborhood). Figure 4C shows that NOBLE accurately captures neuronal dynamics when interpolating within the latent space, as it produces a distribution of voltage responses consistent with those of the previously unseen $\text{HoF}_k^{test}$. In addition, Figure 4A shows that the F-I curves generated by NOBLE remain biophysically meaningful and closely aligned with experimentally observed neuronal behavior.

On the other hand, the parameterizations of the HoF models obtained using multi-objective evolutionary algorithms lack a consistent structure that would enable meaningful interpolation to discover new bio-realistic models. This is illustrated in Figure 4B, where the prediction made by interpolating in between PDE parameterizations deviates significantly from experimental data. This can also be observed in Figure 4A, where the F-I curves obtained by numerically solving PDE models constructed from slightly perturbed parameterizations of $\text{HoF}_k^{test}$ deviate markedly from experimental data.

Interpolating within the latent embedding space enables NOBLE to efficiently generate novel neuron models at scale, while providing up to $4{,}200\times$ faster model predictions than the numerical solver (see Appendix E). Yet, constructing the initial set of bio-realistic HoF models remains time-consuming, prompting the question of how much diversity is truly needed for robust generalization. To examine this, we varied the number of HoF models included in $\{\text{HoF}^{train}\}$ while keeping the dataset size fixed. Performance on voltage traces remained largely stable, but spike-related features degraded markedly as model diversity decreased, highlighting the importance of experiencing sufficient biophysical variability during training. Further details of this analysis are provided in Appendix D.6.

### 4.4 Ensemble Predictions

We now examine how the single trained NOBLE can be used for ensemble predictions. Given a current injection, we run 50 inferences in parallel of NOBLE for the $\{\text{HoF}^{train}\}$ models to produce 50 somatic responses. In Figure 5A (left, middle), we compare these 50 predictions with the corresponding 50 numerical solver simulations from $\{\text{HoF}^{train}\}$. We see that the distribution of curves is very similar.

Since NOBLE enables interpolation between the HoF models used for training, it can generate novel bio-realistic neuron models and produce voltage responses for any neuron model whose latent space representation lies within the convex hull $\mathcal{CH}_{train}$ of the training set $\{\text{HoF}^{train}\}$. We demonstrate this by querying 200 novel models whose features are sampled randomly within $\mathcal{CH}_{train}$. Figure 5A (right) shows that the distribution of curves remains very similar, but the additional samples provide a denser coverage of the response space while maintaining bio-realism, with no artifacts or implausible predictions. We include additional ensemble analyses in Appendix D.5. There, we first validate NOBLE on the test models $\{\text{HoF}^{test}\}$, where predicted and ground-truth responses again show close agreement (Figure 12). Second, we examine local perturbations in the latent embedding space by sampling 50 synthetic models from a small circle around a held-out test model (Figure 11). The resulting responses (Figure 13) resemble small perturbations around the unseen ground-truth trace, demonstrating NOBLE's ability to generalize to unseen models and the smoothness of the latent space.

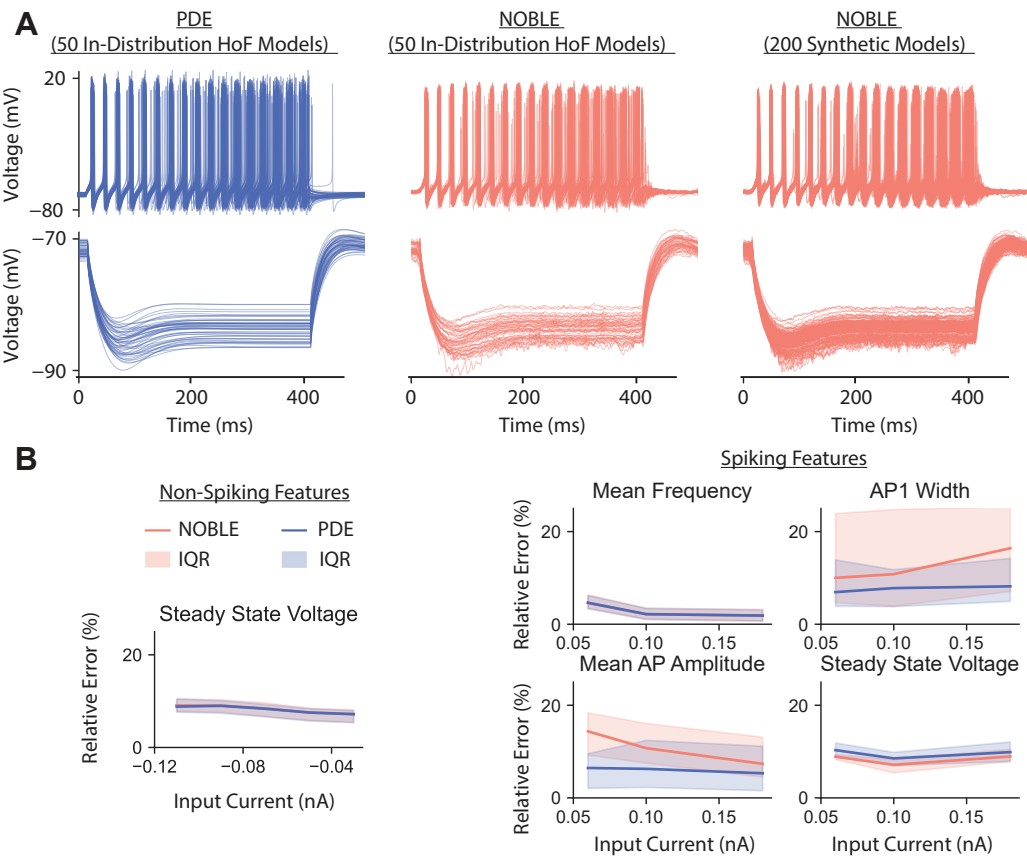

Figure 5: **A**) Distributions of somatic voltage traces across HoF and synthetic models for current injections of $0.1\mathrm{nA}$ (top) and $-0.11\mathrm{nA}$ (bottom). **B**) Relative errors of ensemble predictions from PDE simulations and NOBLE models on $\{\mathrm{HoF}^{train}\}$ compared to experimental data across key features.

Figure 5B shows that NOBLE's ensemble predictions for the full collection of HoF models achieve comparable accuracy to the HoF models themselves when evaluated on the four key electrophysiological features relative to experimental data. Here, we report the mean_frequency, representing the average firing rate, to account for the difference in stimulus activation durations between the experimental recordings and NOBLE, since spikecount is a nonlinear function of time and cannot be directly rescaled. These results demonstrate NOBLE's ability to faithfully represent a diverse set of bio-realistic neuron models while enabling dense interpolation across the model space. However, experimental recordings for this neuron are limited to a single trace across nine amplitudes. Consequently, our comparison is against a single realization rather than a distribution. In practice, experimental features exhibit trial-to-trial variability, which NOBLE is designed to capture, but cannot be directly validated here due to limited experimental data availability.

### 4.5 Choice of Biologically-Informed Latent Embeddings

Feature embeddings are central to NOBLE as they enable generalization across and in-between biological neuron models. Here, we have chosen $I_{thr}$ and $s_{thr}$ for their strong biological interpretability, as a natural 2D representation of firing and non-firing dynamics. To quantify the importance of these embeddings, we conduct an ablation study evaluating NOBLE with lower-dimensional embeddings, as well as without embeddings. We also consider a higher-dimensional embedding that includes AHP_depth, selected for its large variation across intracell HoF models and its low correlation with $I_{thr}$ and $s_{thr}$. Results are reported in Table 1 in terms of the relative L2 error of predicted voltage traces and the four key electrophysiological features on the test set with $\{\mathrm{HoF}^{train}\}$ models.

Table 1: Relative L2 error of `NOBLE` on voltage traces and the four key features, when trained with different embedded features. Results are evaluated on the test set with $\{\text{HoF}^{train}\}$ models.

| Features embedded | Voltage | Steady state voltage | Spikecount | AP1 width | Mean AP amplitude |
|---|---|---|---|---|---|
| None | 12.1% | 1.31% | Never fires | Never fires | Never fires |
| $s_{thr}$ | 2.83% | 1.33% | 4.9% | 233% | 13% |
| $I_{thr}$ | 2.73% | 1.20% | 4.4% | 107% | 14% |
| $s_{thr}, I_{thr}$ | **1.92%** | **1.02%** | **3.1%** | 27% | **8.9%** |
| $s_{thr}, I_{thr}$, `AHP_depth` | 2.16% | 1.04% | 3.3% | **22%** | 9.5% |

Without embeddings, `NOBLE` fails to predict any firing responses, indicating that embeddings are necessary to capture spiking behavior. With a single embedded feature, either $s_{thr}$ or $I_{thr}$, `NOBLE` achieves similar accuracy on voltage traces and `steady_state_voltage`, but errors on spike-related features remain large, with $I_{thr}$ providing better accuracy for `AP1_width`. As discussed earlier, this feature is particularly sensitive to small misalignments in spike peak and after-hyperpolarization minima, which makes its relative error less robust as a metric. Embedding both $s_{thr}$ and $I_{thr}$ yields the best overall performance across all features as well as the voltage trace. Extending the embedding space with `AHP_depth` slightly improves `AP1_width`, but reduces accuracy for the other features. These results show that feature embeddings are important for `NOBLE` to capture both firing and non-firing dynamics, and more broadly for its ability to generalize across diverse neuron models.

While we only considered constructing the latent space from electrophysiological features, `NOBLE` can readily incorporate additional modalities such as gene expression profiles from patch-sequencing data [57]. Learning joint embeddings across modalities could yield a unified latent space linking gene expression, electrophysiology, and morphology, providing a means to test hypotheses that are infeasible experimentally, such as how genes associated with neurological diseases [50] influence neuronal dynamics. In doing so, `NOBLE` would pave the way for improved statistical analysis, more reliable uncertainty quantification, and robust predictive modeling of neuronal behavior.

### 4.6 Feature Specific Physics-Informed Fine-Tuning of `NOBLE`

To preserve overall neural dynamics while improving accuracy on a single specific feature, `NOBLE` can be fine-tuned with a weighted composite loss $\mathcal{L}(\lambda) = \mathcal{L}_{\text{data}} + \lambda \mathcal{L}_F$, where $\mathcal{L}_F$ penalizes deviations in feature $F$, and $\lambda$ controls its influence. We illustrate this by fine-tuning `NOBLE` to improve `sag_amplitude` accuracy, a feature reflecting the hyperpolarization-activated cation channel (Ih). In the human cortex, Ih expression varies with cortical depth, making `sag_amplitude` a relevant physiological marker. We fine-tune on 19,600 subthreshold stimuli with negative amplitudes. Even without a feature-specific loss, fine-tuning reduces the L2 feature error from 70% to 19.2%, and incorporating the feature loss $\mathcal{L}_F$ further lowers it to 9.6% while preserving overall signal fidelity. These results show that `NOBLE` can be selectively refined to prioritize biophysical features of interest without compromising overall performance. Further details are provided in Appendix D.7.

## 5 Conclusion

We introduced `NOBLE`, a neural operator framework for learning the nonlinear somatic dynamics across a population of HoF models for a single neuron. Rather than training separate surrogates for each bio-realistic model, `NOBLE` learns a single neural operator that captures the inherent variability observed in experimental neuron recordings by mapping biologically interpretable embeddings to voltage responses from current injections. Demonstrated on PVALB and VIP neurons, `NOBLE` correctly captured the diverse neuron dynamics with a 4200× speedup over traditional solvers, while maintaining accuracy across key electrophysiological features. Importantly, our work is among the first to benchmark a deep learning based method for predicting membrane potential responses to intracellular current injections against experimental data from the human cortex. `NOBLE` also allows for generating novel, bio-realistic neuron models through interpolation in the latent space, which is not feasible with HoF models. This allows for realistic and efficient ensemble predictions beyond the original set of HoF models. `NOBLE`'s interpretable latent space also offers new insights into how neuron characteristics affect neuron dynamics. `NOBLE` opens a pathway toward modeling larger-scale brain circuits and leveraging multimodal latent spaces to determine relationships between gene expression, electrophysiology, and morphology, as discussed in Appendix F.

## Acknowledgements

L.G. was responsible for the complete technical implementation of this work. C.A.A. and A.A. conceptualized this work. L.G., V.D., and B.T. jointly developed the methodology of the novel `NOBLE` framework. P.H.W. and C.A.A. provided neuroscience-specific domain expertise, contextualizing the relevance and impact of this work within the broader field of neuroscience. P.H.W. supplied the biophysical PDE models and developed the multi-objective optimization pipeline. L.G. and P.H.W. produced the figures. L.G., V.D., B.T., P.H.W., and C.A.A. co-wrote the manuscript. C.A.A. and A.A. provided supervision and editorial comments.

## Funding

A.A. is supported by the Bren Endowed Chair, ONR (MURI grant N00014-23-1-2654), and the AI2050 Senior Fellow program at Schmidt Sciences. C.A.A. is supported by the National Institutes of Health R01 - NS120300 and R01 - NS130126. P.H.W. is supported by the National Institutes of Health R01 - NS130126.

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

## A    Related Works in Machine Learning for Single-Cell Electrophysiology

Earlier applications of machine learning to single-cell electrophysiology focused on directly learning the dynamics of canonical point-neuron models like FitzHugh-Nagumo [31, 32] and Hodgkin-Huxley [33]. Fully connected and convolutional neural networks were trained to reproduce FitzHugh-Nagumo dynamics [34], while ResNet-based multilayer perceptrons showed promise in learning Hodgkin-Huxley dynamics [35, 36]. Physics-informed neural networks (PINNs) extend conventional neural networks by introducing prior knowledge about the underlying dynamical system [37–39]. This formulation was used to predict FitzHugh-Nagumo dynamics [40] and to learn the Hodgkin-Huxley model ionic conductances from simulated voltage recordings [41]. Further refinements with PINNs incorporated wavelet bases to capture localized multiscale dynamics and compute derivatives analytically, improving both accuracy and convergence speed when training on the FitzHugh-Nagumo model [42]. While these methods demonstrate capabilities in capturing the core spiking dynamics of these simplified models, their ability to accurately represent the full spectrum of electrophysiological behavior, particularly the highly nonlinear onset of firing, remains largely untested. Moreover, as function approximators, they necessitate retraining for each new input stimulus, significantly limiting their practical utility.

To address some of these limitations, Centofanti et al. [43] explored using operator learning approaches for forward simulations of the Hodgkin-Huxley model. Among other approaches, FNOs showed promising results by demonstrating a strong capacity for learning the governing operator of this biophysical system. However, this work still exhibits key limitations and a limited scope: (1) it relies on relatively simple simulated data from a point-neuron model, (2) it does not explicitly attempt to capture the full spectrum of electrophysiological dynamics, particularly the highly nonlinear onset of firing, and (3) its formulation on a single operator inherently lacks the capacity to represent the trial-to-trial variability observed in biological recordings. Related work such as NeuPRINT [44] also leverages deep learning to capture trial-to-trial variability. However, it operates on slower in vivo 2-photon calcium imaging data and models population-level fluorescence dynamics, rather than the fast intracellular voltage dynamics of individual neurons.

## B    Electrophysiological Features

For electrophysiological feature extraction and metrics, we use code from the Electrophys Feature Extraction Library (eFEL) available at

https://github.com/BlueBrain/eFEL

The formulas, codes, and more details about each electrophysiological feature can be found at

https://efel.readthedocs.io/en/latest/eFeatures.html

We list below 16 important electrophysiological features and metrics of interest when constructing neuron models (where AP denotes action potential and AHP denotes after-hyperpolarization):

- `AHP_depth`: Relative voltage values at the first AHP

- `AHP_time_from_peak`: Time between AP peaks and first AHP depth

- `AHP1_depth_from_peak`: Voltage difference between the first AP peak and first AHP depth

- `AP1_peak`: The peak voltage of the first AP

- `AP1_width`: Width of first spike at half spike amplitude, with the spike amplitude taken as the difference between the minimum between two peaks and the next peak

- `decay_time_constant_after_stim`: The decay time constant of the voltage right after the stimulus

- `depol_block`: Check for a depolarization block. Returns 1 if there is a depolarization block or a hyperpolarization block, and returns 0 otherwise.

- `inv_first_ISI`: 1.0 over first interspike interval; returns 0 when no interspike interval

- `mean_AP_amplitude`: The mean of all of the AP amplitudes

- `mean_frequency`: The mean frequency of the firing rate

- `sag_amplitude`: The difference between the minimal voltage and the steady state at the end of the stimulus

- `spikecount`: Number of spikes in the trace, including outside of stimulus interval

- `steady_state_voltage`: The average voltage after the stimulus

- `steady_state_voltage_stimend`: The average voltage during the last 10% of the stimulus duration.

- `time_to_first_spike`: Time from the start of the stimulus to the maximum of the first peak

- `voltage_base`: The average voltage during the last 10% of time before the stimulus

# C  Method

## C.1  Impact of Subsampling on Neuron Features

We present the results of the analysis conducted to determine the maximum subsampling factor that preserves the fidelity of extracted neuron features, mentioned in Section 3.1. The results are displayed in Figures 6 and 7 for the low-pass filtering followed by decimation in time subsampling strategy.

We first computed the relative error between the raw, non-subsampled HoF model voltage responses and the experimental data across all amplitudes. For each amplitude, we identified the minimum relative error across all HoF models, and then aggregated these minima to compute the mean and standard deviation. These serve as a reference for the inherent *worst-case* discrepancy between simulations and experimental recordings in the absence of any subsampling. We visualize the mean as a solid black line and the standard deviation as dotted black lines.

Next, we repeated a similar analysis to quantify the additional relative errors introduced by subsampling. For each subsampling factor, we calculated the relative error between the subsampled and original HoF responses for each amplitude. These errors were then averaged across all HoF models, and the distribution of these averages is summarized using the mean (solid line), standard deviation (shaded region), and min/max (error bars).

This study shows that for most electrophysiological features, subsampling introduces negligible additional error. The most sensitive features were `AP1_Width` and `AP1_Peak`, which exhibited noticeable deviations at higher subsampling rates. To ensure we remain within the bounds of the intrinsic simulation-experiment discrepancy, we adopt a conservative downsampling factor of $3\times$, which maintains fidelity while reducing computational load.

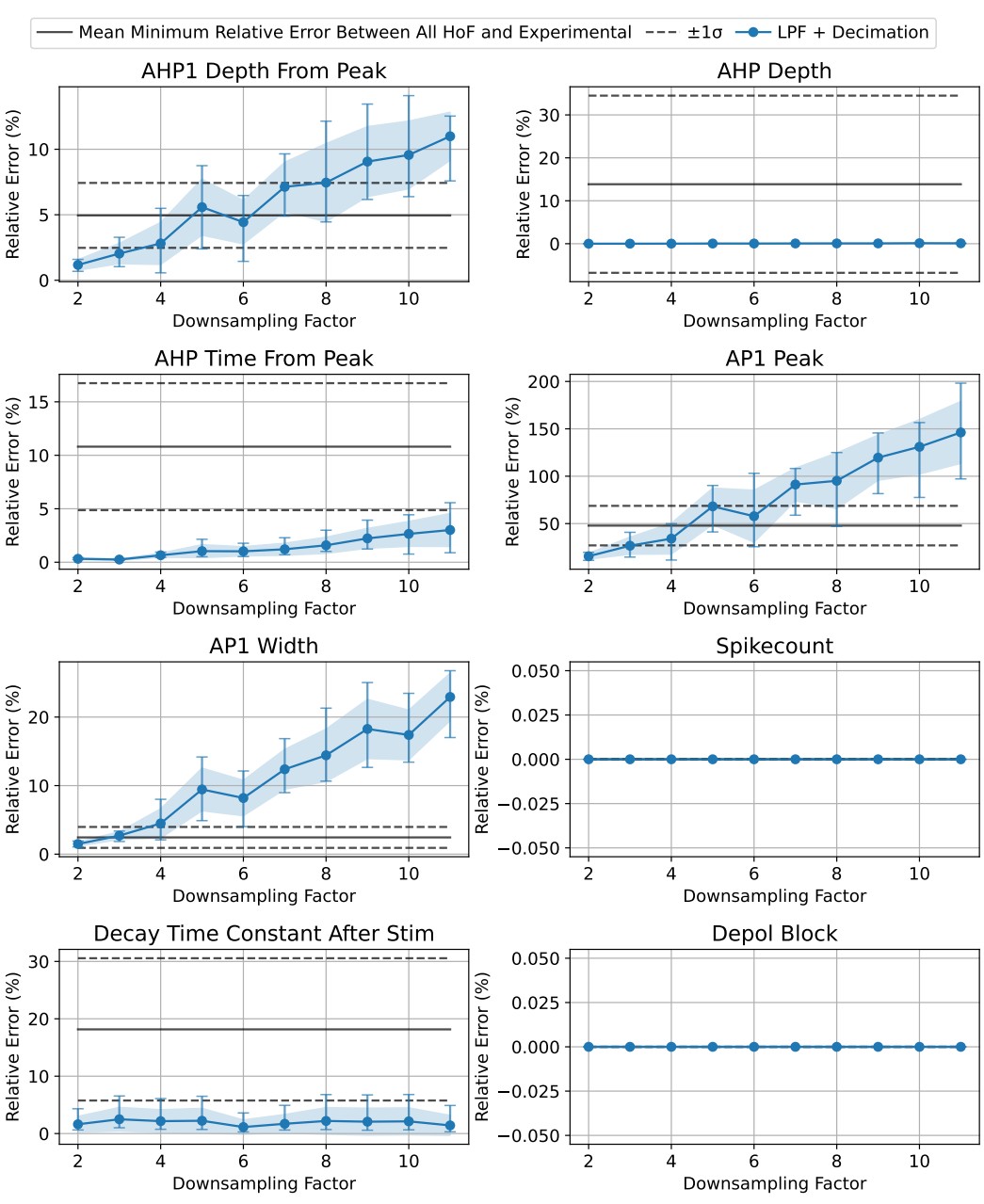

Figure 6: Analysis of the relative errors introduced in neuron feature computation as a function of subsampling factor, using low-pass filtering followed by decimation in time. The solid and dotted black lines indicate the mean and standard deviation, respectively, of the minimum relative error between non-subsampled HoF and experimental responses. The solid blue line, shaded region, and error bars represent the mean, standard deviation, and minimum–maximum statistics of the relative error between non-subsampled and subsampled HoF responses.

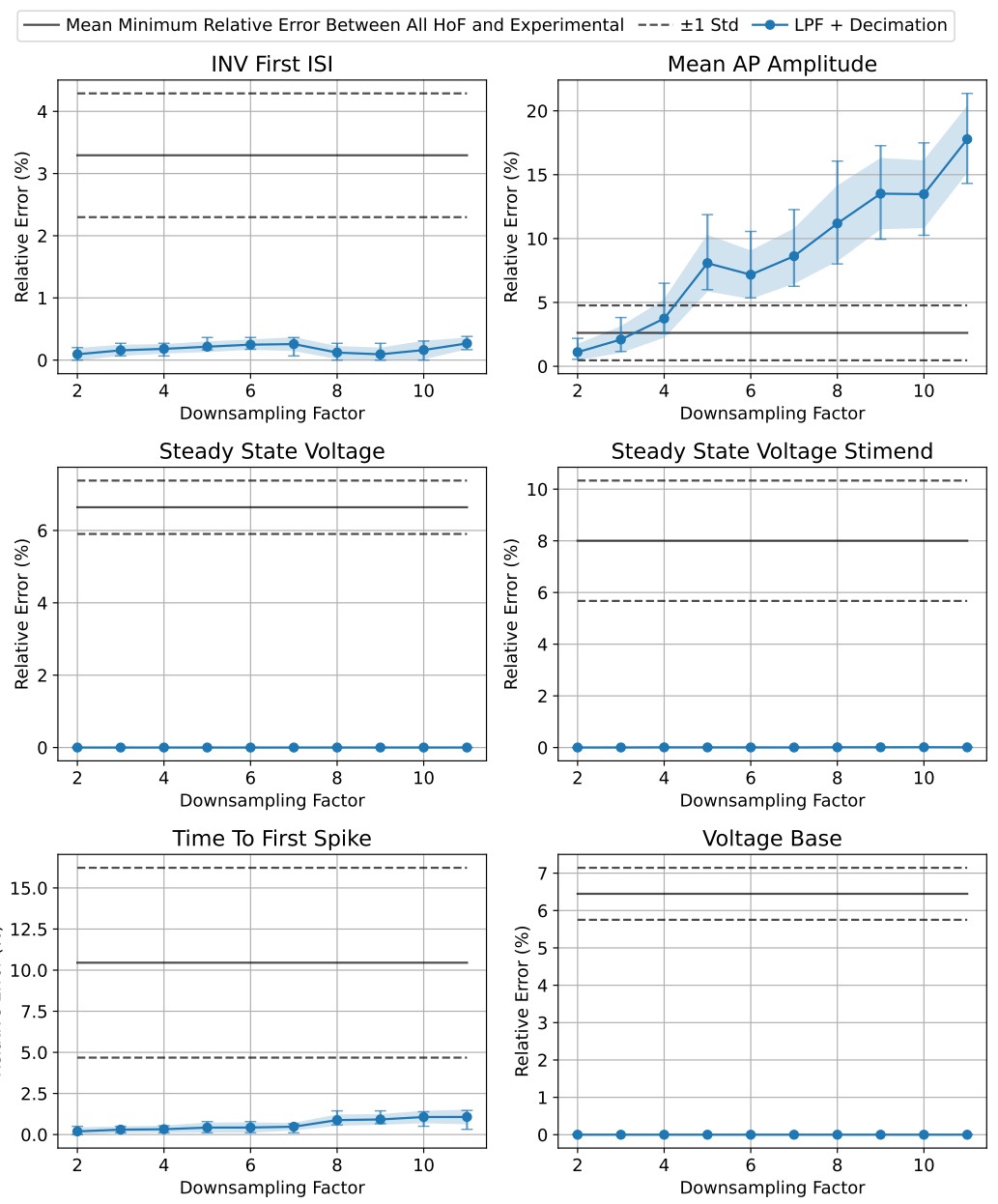

Figure 7: Analysis of the relative errors introduced in neuron feature computation as a function of subsampling factor, using low-pass filtering followed by decimation in time. The solid and dotted black lines indicate the mean and standard deviation, respectively, of the minimum relative error between non-subsampled HoF and experimental responses. The solid blue line, shaded region, and error bars represent the mean, standard deviation, and minimum–maximum statistics of the relative error between non-subsampled and subsampled HoF responses.

## C.2 Input Current Amplitude Distribution

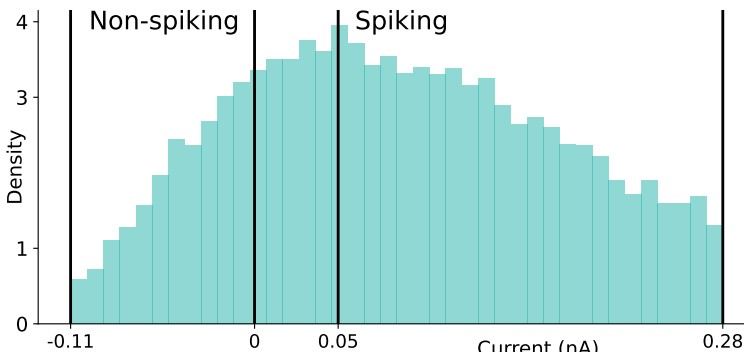

Figure 8: Distribution of square-pulse amplitudes in $[-0.11, 0.28]$nA considered. There is a spiking threshold (between 0 to 0.05nA) where neuron responses transition from being non-spiking to spiking.

## C.3 The Fourier Neural Operator Architecture

Neural operators [25–27] are a principled way to generalize neural networks to learn operators mapping functions to functions, with a universal operator approximation property [28]. Neural operators compose linear integral operators $\mathcal{K}$ with pointwise nonlinear activation functions $\sigma$ to approximate highly nonlinear operators. More precisely, we define the neural operator

$$\mathcal{G}_\theta = \mathcal{Q} \circ \sigma(W_L + \mathcal{K}_L + b_L) \circ \cdots \circ \sigma(W_1 + \mathcal{K}_1 + b_1) \circ \mathcal{P} \tag{2}$$

where $\mathcal{P}, \mathcal{Q}$ are the pointwise neural networks that encode the lower dimension function into a higher-dimensional space and vice versa. The model stacks $L$ layers of $\sigma(W_l + \mathcal{K}_l + b_l)$ where $W_l$ are pointwise linear operators (matrices), $\mathcal{K}_l$ are integral kernel operators, $b_l$ are bias terms, and $\sigma$ are fixed activation functions. The parameters $\theta$ consists of all the parameters in $\mathcal{P}, \mathcal{Q}, W_l, \mathcal{K}_l$ and $b_l$. Kossaifi et al. [56] maintain a comprehensive open-source PyTorch library for learning neural operators, which serves as the foundation for our implementation. Prior knowledge of the relevant physics laws and differential equations can also be incorporated as additional loss terms during training, to supplement or replace reference data, as done with physics-informed neural operators [58–60].

A variety of neural operators have been proposed, such as the Fourier Neural Operator (FNO) [29, 30], and successfully applied to a wide range of problems [61–63]. A FNO is a neural operator using Fourier integral operator layers, which are defined via

$$\big(\mathcal{K}(\phi)v_t\big)(x) = \mathcal{F}^{-1}\Big(R_\phi \cdot (\mathcal{F}v_t)\Big)(x) \tag{3}$$

where $R_\phi$ is the Fourier transform of a periodic function $\kappa$ parameterized by $\phi$. On a uniform mesh, the Fourier transform $\mathcal{F}$ can be implemented using the fast Fourier transform (FFT).

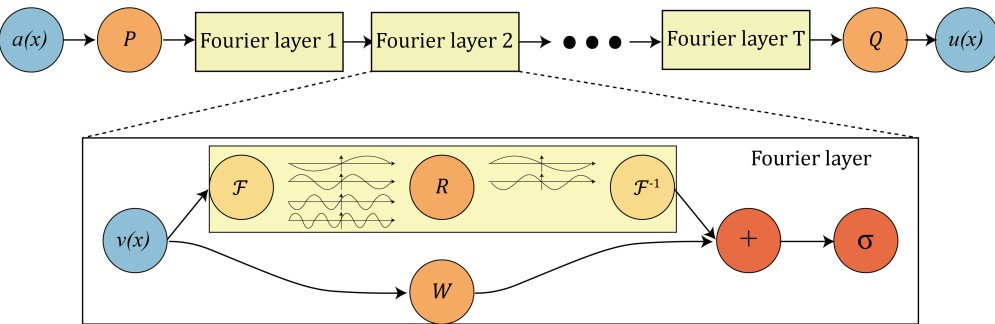

Figure 9: The Fourier Neural Operator (FNO) architecture (extracted from [29]).

## D   Experiments

### D.1   Dataset

We have access to 60 HoF models {HoF} obtained using a multi-objective evolutionary optimization strategy. We use 50 of them during training, {HoF$^{train}$}, and keep the remaining 10 {HoF$^{test}$} for testing.

Figure 10 displays the latent representations in normalized $(I_{thr}, s_{thr})$-space of these HoF models.

The training dataset is composed of 75,600 samples, where the current injections are sampled as described in Section 3.2, each of which is associated randomly to one of {HoF$^{train}$}. The samples are generated using a numerical solver [45] built on the NEURON simulation environment [46].

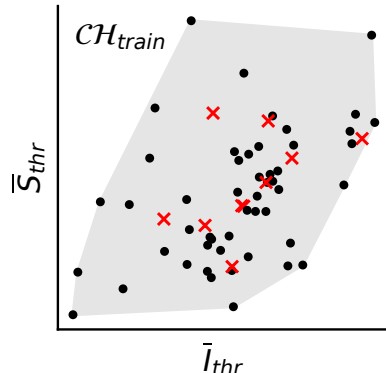

Figure 10: Latent representations of neuron models in the normalized $(I_{thr}, s_{thr})$-space. Black dots indicate the 50 training HoF models {HoF$^{train}$}, and red crosses the 10 test HoF models {HoF$^{test}$} excluded during training.

### D.2   Evaluation and Evaluation Metrics

NOBLE is trained using the relative L4 error, computed via

$$\text{Relative Lp error}(x, y\,;\epsilon) = \frac{\|x - y\|_p}{\|y\|_p + \epsilon} = \frac{(\sum_i |x_i - y_i|^p)^{1/p}}{(\sum_i |y_i|^p)^{1/p} + \epsilon}. \tag{4}$$

The relative L4 error was selected as the training loss after a preliminary training study on a small dataset, where it consistently preserved spike-related features, especially amplitudes and widths, more effectively than the relative L2 error. While this choice ensured the model captured the electrophysiological details most relevant to our setting, NOBLE is compatible with any other loss function that may be more suitable in different contexts.

Although NOBLE is trained to minimize the relative L4 error, we report results using the relative L2 error, as it provides a more common and interpretable measure of accuracy. We first evaluate performance on voltage traces. Note that even small temporal shifts between predicted and ground-truth voltage responses can result in large relative errors. To better evaluate physiologically meaningful behavior, we also report errors on four key electrophysiological features:

- spikecount: number of spikes in the trace
- AP1_width: width of the first spike at half amplitude
- mean_AP_amplitude: mean amplitude of all action potentials
- steady_state_voltage: average voltage after the stimulus

For benchmarking, we compare NOBLE predictions against numerical simulations obtained from HoF models and experimental data. HoF models were optimized to produce the closest approximations to experimental recordings and capture the biological variability required for a meaningful benchmark. In contrast, existing machine learning methods, such as the ones discussed in the introduction, are not designed to reproduce such variability and are therefore unsuitable as baselines. Furthermore, since the PDE solvers used to generate the dataset are themselves approximations with non-negligible error, driving prediction error below the solver–experiment gap risks overfitting to the solver rather than improving alignment with real recordings. Therefore, we tuned hyperparameters only up to the solver–experiment error level, as marginal gains on PDE data are unlikely to yield meaningful improvements relative to experimental recordings.

### D.3 Testing on VIP Neuron HoF Models Included in the Training Set

To assess NOBLE 's ability to perform well on different neuron models, we trained it on a VIP neuron using the same architecture as in the PVALB case with 1.8M trainable parameters. The model was optimized for 450 epochs with the Adam optimizer with learning rate 0.004 and the ReduceLROnPlateau scheduler with factor 0.8 and patience 4, minimizing the relative L4 error. For the embeddings, the neural operator in NOBLE takes the stacked embeddings of the normalized model features $I_{thr}$ and $s_{thr}$ associated with HoF$_\ell$ (with $K = 1$ frequency) and $I$ (with $K = 11$ different frequencies).

The trained model achieves a relative L2 error of 2.5% on voltage traces and relative L2 errors of 9.0% for spikecount, 20% for AP1_width, 10% for mean_AP_amplitude, and 0.99% for steady_state_voltage.

These results are comparable to those obtained for PVALB, indicating that NOBLE, with the same latent embedding space, also performs well when trained on different neuron types.

### D.4 Neighborhood Considered in Interpolation Experiments

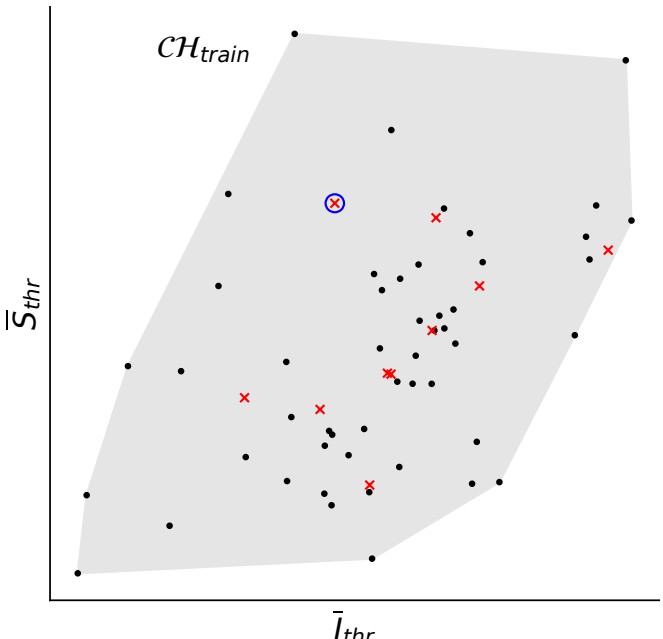

Figure 11: Latent representations in normalized $(I_{thr}, s_{thr})$-space of $\{\text{HoF}^{train}\}$ (black dots) and $\{\text{HoF}^{test}\}$ (red crosses) models. The latter lie in the convex hull $\mathcal{CH}_{train}$ of the $\{\text{HoF}^{train}\}$ models. In the interpolation experiment of Section 4.3, we construct a small neighborhood around a given HoF$_k^{test}$ that defines a region of latent space not encountered during training, and sample 50 unseen models from this neighborhood. The boundary of this neighborhood is shown (blue circle) for an example HoF$_k^{test}$ model.

## D.5 Auxiliary Ensemble Prediction Figures

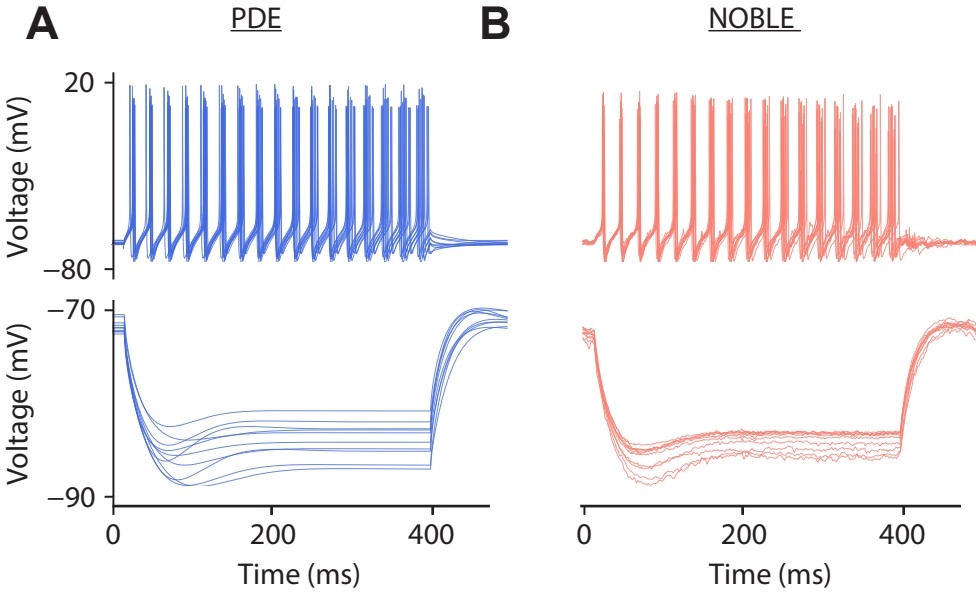

Figure 12: Distribution of somatic voltage traces across HoF models $\{\text{HoF}^{test}\}$ for current injections of $0.1$nA (top row) and $-0.11$nA (bottom row). **A**) Ground truth voltage responses from HoF simulations, **B**) NOBLE predictions.

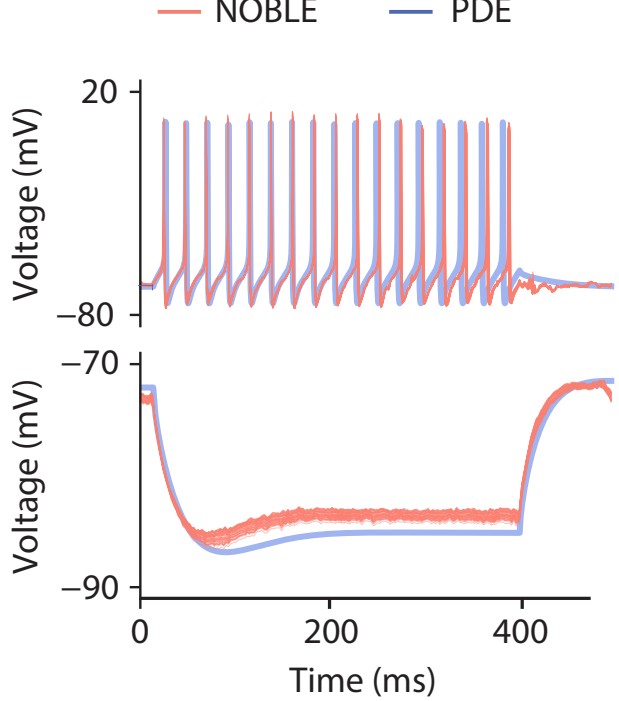

Figure 13: Distribution of somatic voltage traces across $50$ synthetic HoF models sampled from a small circle centered on a HoF model in $\{\text{HoF}^{test}\}$ not experienced during training, as shown in Figure 11. Results are shown for current injections of $0.1$nA (top) and $-0.11$nA (bottom). The ground truth voltage response from the HoF simulation not experienced during training is shown in blue, and the $50$ synthetic NOBLE predictions in orange.

## D.6 On the HoF Models Availability Requirement

Training `NOBLE` on 75,600 samples required approximately 4 days on a 64GB NVIDIA Tesla P100 GPU (300 epochs). Once trained, for a given input current, `NOBLE` can synthesize arbitrarily many voltage responses of synthetic HoF models almost instantaneously by interpolating within $\mathcal{CH}_{train}$ (Figure 10). Thus, `NOBLE` amortizes the high upfront cost of bio-realistic neuron model generation and enables scalable response synthesis at negligible inference cost. To investigate how much model diversity is needed during training for effective amortization, we varied the number of HoF models used to construct the training set $\{\text{HoF}^{train}\}$ while keeping the total number of samples fixed. Results are summarized in Table 2 in terms of the relative L2 error of predicted voltage traces and the four key electrophysiological features when evaluated on the $\{\text{HoF}^{test}\}$ models.

Table 2: Predictive performance of `NOBLE` on voltage traces and the four key electrophysiological features using the relative L2 error metric when the training set $\{\text{HoF}^{train}\}$ is constructed by including varying numbers of models. Results are evaluated on $\{\text{HoF}^{test}\}$.

| #HoFs included | Voltage | Steady state voltage | Spikecount | AP1 width | Mean AP amplitude |
|---|---|---|---|---|---|
| 50 | 11.7% | 2.0% | **9.2%** | **350%** | **14%** |
| 40 | 10.9% | **1.8%** | 19% | 920% | 17% |
| 30 | 10.9% | 1.9% | 20% | 3004% | 20% |
| 20 | **10.6%** | 1.9% | 45% | 1698% | 20% |

The relative L2 error on voltage traces is largely insensitive to HoF diversity, whereas spike-related features, particularly `spikecount` and `AP1_width`, degrade substantially as diversity decreases. As discussed earlier, `AP1_width` is particularly sensitive to small misalignments in spike peak and after-hyperpolarization minima, which can shift the half-voltage reference and lead to large relative errors even when the underlying traces are close. This makes `AP1_width` less reliable for direct comparison than other features. Overall, these findings indicate that while `NOBLE` amortizes the cost of HoF generation, effective bio-realistic synthesis still requires sufficient model diversity to learn robust representations of neural dynamics. Depending on the purpose of the study and the solver–experiment error gap, the number of HoF models required for training can be adjusted accordingly. Moreover, for any given electrophysiological feature of particular interest, further fine-tuning can be used to refine predictions and improve generalization, as we discuss next.

## D.7 Additional Information on Fine-Tuning

Suppose the objective is to capture overall neural dynamics while placing particular emphasis on one specific electrophysiological feature. In this setting, the loss function can be designed to prioritize the feature of interest. Let $F$ denote a feature computed on the ground-truth signal and $\hat{F}$ the corresponding feature computed from `NOBLE`'s output. A feature-specific loss can then be defined as $\mathcal{L}_F = \|F - \hat{F}\|$, which directly penalizes deviations in the feature of interest.

To illustrate this, we fine-tune the previously trained `NOBLE` to enhance `sag_amplitude` predictive performance. This feature is particularly relevant as it reflects the presence of the hyperpolarization-activated cation channel (Ih). In human cortex, the expression of Ih varies with cortical depth, making `sag_amplitude` a useful marker. Although the broader role of Ih in shaping neuronal and network properties is not yet fully understood, it is thought to regulate neural excitability and coincidence detection.

We start from a pretrained `NOBLE` model and further optimize the weights of the neural operator using the feature-specific loss. Relying solely on a feature-specific loss, such as $\mathcal{L}_{sag}$, risks causing `NOBLE` to overfit to `sag_amplitude`, improving that metric while degrading performance on other features and overall voltage trace fidelity. To mitigate this issue, one possible strategy is to introduce an anchor loss, as proposed in PINO [58], which penalizes deviations from the pretrained operator during fine-tuning. Combining the anchor loss with $\mathcal{L}_{sag}$ could constrain optimization so that gains on a single feature do not come at the expense of overall signal fidelity, thereby encouraging balanced gains across all electrophysiological features.

Another approach, which we adopt, is to define a composite loss that encourages further accuracy on the voltage traces while prioritizing the feature of interest, `sag_amplitude`:

$$\mathcal{L}(\lambda) = \mathcal{L}_{\text{data}} + \lambda\mathcal{L}_{\text{sag}}.$$

To evaluate this approach, we construct a smaller dataset of 19,600 stimulus waveforms with negative non-zero amplitudes, since this regime elicits non-firing responses for which the `sag_amplitude` can be reliably computed. We then fine-tune the pretrained `NOBLE` model by minimizing the relative L4 error for $\mathcal{L}_{\text{data}}$ and the relative L2 error for $\mathcal{L}_{\text{sag}}$, using the Adam optimizer with learning rate of $0.0005$, and the ReduceLROnPlateau scheduler with factor $0.4$ and patience $12$. The results, reported in Table 3, summarize the relative L2 error of predicted voltage traces and `sag_amplitude` on the test set of $\{\text{HoF}^{train}\}$.

Table 3: Predictive performance of `NOBLE` fine-tuned on `sag_amplitude`. Metrics are reported as relative L2 errors on voltage traces and on `sag_amplitude`, with the training set $\{\text{HoF}^{train}\}$ constrained to samples with negative non-zero amplitudes. Results are evaluated on the test set of $\{\text{HoF}^{train}\}$. Here, $\lambda$ denotes the weighting factor of the feature-specific loss in the composite loss.

| | **Before optimization** | | **Epoch** 100 | | **Epoch** 200 | | **Epoch** 300 | |
|---|---|---|---|---|---|---|---|---|
| $\lambda$ | **Voltage** | **Sag amplitude** | **Voltage** | **Sag amplitude** | **Voltage** | **Sag amplitude** | **Voltage** | **Sag amplitude** |
| 0 | 0.14% | 69.6% | 0.064% | 22.2% | **0.041%** | 20.3% | 0.043% | 19.2% |
| 25 | 0.14% | 69.6% | **0.055%** | **13.2%** | 0.047% | **12.2%** | **0.035%** | **9.6%** |

Even without an additional `sag_amplitude` loss, fine-tuning on the restricted non-firing regime of the dataset improves both trace prediction and `sag_amplitude` performance, since in this setting every sample is subthreshold and the feature can be computed consistently.

Including the feature-specific loss provides a further improvement of approximately 10% on `sag_amplitude`, while maintaining overall signal fidelity. Note that both settings converged after 300 epochs, ensuring that the comparison is fair.

These results demonstrate that prioritizing a feature through the loss function can yield targeted improvements without sacrificing global accuracy.

# E    Comparison of Sample Generation Time

The trained `NOBLE` generates predictions significantly faster than the reference numerical solver from [45].

We record the time necessary to generate 10,000 predictions on a workstation equipped with a single NVIDIA RTX 4090 GPU (24GB VRAM), an AMD Ryzen 9 7900X CPU, and 64GB of system RAM. The numerical solver only generates a single prediction at a time and takes roughly 36,200 seconds to generate 10,000 predictions. When doing one inference at a time with `NOBLE` (i.e. batch size of 1), we generate 10,000 predictions in 157 seconds, i.e. a speedup of approximately 230× compared to the solver.

In addition, `NOBLE` can easily be accelerated on a single GPU by generating multiple predictions at the same time. In particular, with a batch size of 1000, `NOBLE` generates 10,000 predictions in 8.59 seconds, i.e. a speedup of approximately 4200× compared to the solver.

# F   Scope and Future Directions

`NOBLE` successfully captures the dynamics across HoF models but is currently limited to single-neuron settings. A natural next step is to extend `NOBLE` to multi-neuron configurations with time-varying stimuli, enabling applications such as neuron classification and predicting multi-neuron dynamics. The embedding space used in our experiments is low-dimensional, constructed from two interpretable features derived from a biological neuron model's F-I curve. Extending this to a learnable, higher-dimensional continuous embedding space represents a promising direction for future work. Beyond this, `NOBLE` could also integrate additional modalities such as gene expression or morphology to build a unified latent space linking molecular, electrophysiological, and structural characteristics, as discussed in the main text.

Although `NOBLE` demonstrates strong performance in modeling nonlinear neuronal dynamics, our study makes a few deliberate scope choices. These do not represent inherent limitations of the framework but rather natural starting points, each of which can be extended with minimal or no modifications.

- **Input currents**: We restricted our attention to square-pulse DC step currents as these are widely used in electrophysiological experiments and represent a common protocol for characterizing neuron behavior. However, the `NOBLE` framework is not specific to these types of input currents: time-varying inputs can be incorporated directly by including examples during training. In such cases, the amplitude embedding can be removed or adapted (e.g., embedding a function of the amplitude, such as a moving-average modulation or the maximum amplitude).

- **Choice of operator learning architecture**: We used the FNO primarily for its computational efficiency and strong generalizability. Moreover, since FNOs use the FFT and thus require inputs and outputs on equidistant grids, they align naturally with our data, where both simulations and experimental recordings are sampled at constant timesteps. For non-uniformly spaced data, alternative neural operators such as geometry-informed neural operators (GINOs) [64, 59] could be used within the same framework.

- **Training cost vs. efficiency**: Training `NOBLE` on 75,600 samples for 300 epochs took approximately four days on a 64GB NVIDIA Tesla P100 GPU, which is small compared to the $\sim$600,000 CPU hours required to generate the HoF models via evolutionary optimization. However, once trained, `NOBLE` enables fast inference and the instantaneous generation of infinitely many bio-realistic voltage traces through latent space interpolation, capabilities not possible with the original PDE models.

- **Neuron populations considered**: We focused on inhibitory neurons (PVALB and VIP), which show strong heterogeneity in morphology, gene expression, and electrophysiology, making them a stringent test for generalization. However, `NOBLE` is not restricted to inhibitory neurons and can naturally extend to excitatory types. It can also be applied to larger populations or multiple neurons within a family by expanding the latent embedding space with additional electrophysiological features that capture intracellular variability.

