# OpenReview forum: "NOBLE - Neural Operator with Biologically-informed Latent Embeddings to Capture Experimental Variability in Biological Neuron Models"
_NeurIPS.cc/2025/Conference — NeurIPS 2025 poster_

### Official Review · Reviewer_d3WE · 2025-06-04

**Clarity:** 2
**Significance:** 3
**Originality:** 3
**Rating:** 5
**Confidence:** 3

**Summary:**

This paper describes a method to generate neural voltage traces starting from easily measurable neuron features (F-I curve). The method allows to sample voltage traces of synthetic neurons that are consistent with electrophysiological data. Using a neural operator approach, the method is insensitive to the discretization/integration method used in numerical solvers. The result is a speedup in generating voltages traces compared to numerical solvers. The technical contribution of the paper is the application of an existing neural operator approach to neural electrophsiological data.

**Questions:**

- I'm not convinced by how the demonstrated use cases of the method enables a "better understanding of cellular composition and computations, neuromorphic architectures, large-scale brain circuits, and general neuroAI applications". The proposed method does apparently not reveal biophysical parameters to identify the neuron types so I'm not sure how it can reveal cellular composition. Its help to inform neuromorphic architectures and large-scale brain circuits is a stretch from the shown results. Figure 7 shows consistency in some electrophysiological features, but this seems somewhat limited towards "predictive modeling of neuron dynamics".
- "It learns a direct mapping from input currents and an interpretable, continuous latent space of neuron features to voltage traces (Figure 1A).": How are the traces "interpretable"? Is there a way to extract the complete biophysical parameters of the models generating the voltage traces?
- "...generate families of models, sometimes referred to as "hall-of-fame" (HoF) models to represent a single cell." The authors don't clearly describe whether (or how) the models are combined. Does this family form a pool from which the best fit is selected, or are they somehow combined (e.g. a linear combination as with basis functions).
- "intrinsic variability" is mentioned throughout the article. Does this term include neuron heterogeneity or specifically refer to variability in the recording of a given neuron? I think it would be helpful to distinguish the two.
- The fourier transform of each embedding dimension yields trivial functions. Couldn't the first fourier transform of FNO and the sine/cosine be trivially discarded?
- what does "Cellular models representing multiple data modalities" mean? Does this refer to a model of the measurement (e.g. LFP, multi unit recording etc?)

Other minor comments
- I'm pretty sure that in "where Rϕ is the Fourier transform of a periodic function", you meant F (R seems to be a linear mixing layer). Same in the appendix.
- The notation \sigma(W+K+b) is a bit misleading. Please describe the meaning of this notation (mathematically)
- If possible, it would be helpful to relate the parameters in eq (1) to figure 1.
- It took me more time that I wanted to understand that eq (2) is not a time series itself, but rather a vector of the time series at a given time point. Perhaps add some notation to clarify this.
- figure 4: I see no reason to separate the plots PDE and Noble. Please superimpose them to better assess the differences (as in figure4)

**Ethical Concerns:**

["NO or VERY MINOR ethics concerns only"]

**Final Justification:**

Thanks for addressing my questions. I think this paper is a useful contribution to the area. I raised my score accordingly.

**Limitations:**

- The limitation to generate voltage traces from a step function is not sufficiently addressed. PDE based approaches would not have this limitation.

**Paper Formatting Concerns:**

No paper formatting concerns

**Quality:**

3

**Strengths And Weaknesses:**

Strengths:
- The generation of voltage traces from an latent embedded space is a neat idea.
- The technical quality of the paper and the positioning relative to neuroscience research is well explained

Weaknesses:
- There are no direct comparisons with other state-of-the-art methods (deep learning or other statistical methods) to generate voltage traces. It is hard to assess the relative quality of the fits without such benchmarks.
- The paper lacks in clarity ( some expressions cannot be deciphered without digging into references), and precision (math is not precisely described.). Comments in Questions below  reflect this lack of clarity.
- The shown input currents appear to be step functions, capturing an arguably limited extent of a neuron's dynamical features. Assuming there is no direct way to relate the mapped voltage traces to the biophysical parameters of the neuron model, the HoF models could not be used to predict the neuron states to dynamic input currents.

---

> ### Author Rebuttal · Authors · 2025-07-31
>
> We thank the reviewer for their thoughtful summary and for recognising the technical quality and relevance of our work within the context of neuroscience research. We are pleased that the reviewer found the use of a latent embedded space for generating voltage traces to be a compelling aspect of our approach, and appreciate the recognition of NOBLE’s ability to produce biophysically plausible outputs efficiently. We have addressed all raised concerns below and hope our response satisfactorily resolves the reviewer’s points. We are happy to provide further elaboration or modifications during the discussion period.
>
> ---
>
> **W1: PDE benchmarks are biologically grounded and suitable for learning**
>
> We understand the concern about limited comparisons to other baselines. Our primary goal is to evaluate whether the neural operator framework can capture the biological variability seen in real-world experiments. To this end, we used data generated from PDE-based models which were selected via evolutionary multi-objective optimization as the best available approximations of experimental recordings, and exhibit the desired variability. Now, other existing ML approaches [28-33;37-40] in neuroscience are not equipped to capture variability, and can therefore not be used for a comparison. While other models could be used within NOBLE and trained on the same data, we believe this would not offer additional insight. Our focus is on learning representations that reliably capture key e-features across neuron dynamics and parameter regimes, and NOBLE already demonstrates strong performance on this benchmark. Since the PDE solvers generating the dataset themselves are approximations of biological reality with non-negligible errors, further reducing prediction errors would lead to further overfitting to the PDE solver simulations but probably not to substantially better-captured dynamics and e-features when compared to true real-life recordings. For this reason, we tuned hyperparameters up to the level of error seen between the PDE models and experimental data, as marginal error improvements on the PDE data are unlikely to translate meaningfully when compared to real experimental data. We included a note on this in Section 4.1 of the revised paper.
>
> ---
>
> **W2: Input Currents Selection**
>
> We agree with the reviewer that richer time-varying stimuli would allow for broader validation. We focused on DC step currents because they are widely used in electrophysiological experiments and represent a common protocol for characterising neuron behaviour. More importantly, they offer a readily available source of experimental data to validate NOBLE’s predictions. That said, extending NOBLE to handle arbitrary time-varying inputs is a natural next step that we intend to explore in future work, and added a comment about this at the end of the paper.
>
> ---
>
> **Q1: Improving biological understanding**
>
> The biologically interpretable embeddings of NOBLE provide a testing ground towards understanding the relationship between neuron properties (electrophysiology, morphology, transcriptomics) and computation. In this first version of NOBLE, we only included 2 prominent e-features in the embedding that is distinct between inhibitory neo-cortical cell-types, but this embedding could incorporate any number of arbitrary features, including morphological features and ion channel gene expression. Through perturbations (most of which are experimentally infeasible and also not possible with biophysical PDE modeling), one could then directly test how these features relate to a neuron’s electrical properties, informing us of how location dependent variations of cellular composition in the human brain might relate to their computations.
>
> Another promising approach involves sampling the embedding space, running NOBLE on each point, and computing e-features to generate heatmaps. This allows to visualise how features vary across the latent space (thereby informing about the relation between the features visualized and those defining the latent space), helps highlight the geometry of the learned representations while also offering new insights into the underlying structure, variability, and diversity of neuronal behaviour. We added an example of such a heatmap and highlighted this as an exciting direction to explore further in future work
>
> While NOBLE does not extend to large scale circuit simulations yet, it addresses current bottlenecks in such modeling:
> 1. The scalable generation of diverse bio-realistic single neuron models
> 2. The computational scalability of running these simulations on large populations of neurons
>
> ---
>
> **Q2: The interpretability lies in the feature-based latent space**
>
> We thank the reviewer for pointing this out. Here, the term “interpretable” refers to the e-features used to embed the models. These features are biophysically meaningful and allow us to condition NOBLE on a feature space that is directly linked to neuronal function (e.g. sthr and Ithr). We clarified this terminology in the revised version.
>
> The voltage traces themselves are not inherently interpretable in the sense of revealing explicit biophysical parameters. However, one could in principle attempt to recover the underlying model parameters (e.g. conductances or membrane properties) using inverse modelling or system identification approaches (e.g. PDE-SINDy)
>
> ---
>
> **Q3: Clarification on the HoF models**
>
> The HoF models are not constructed through combination or linear mixing. They are obtained via a multi-objective evolutionary optimization over a large space of candidate biophysical models (~600,000 PDEs), which aims to identify parameter configurations that best match a set of e-features extracted from experimental recordings. The final HoF set consists of multiple distinct models, each offering a biophysically plausible fit to the experimental data, but differing in specific parameters (e.g. ion channel distributions, membrane properties). These models reflect the trial-to-trial variation of biological systems. We clarified this process in the paper to avoid confusion with methods involving basis function combinations or linear mixtures of models.
>
> ---
>
> **Q4: “Intrinsic variability” refers to repeated responses from the same neuron, not inter-neuron heterogeneity**
>
> We thank the reviewer for bringing this source of confusion to our attention. Here, the term “intrinsic variability” specifically refers to the trial-to-trial variability observed in repeated recordings of a single neuron in response to the same current injection. This type of variability can arise from channel noise, spontaneous fluctuations in cellular state, or small experimental perturbations. We revised the paper to make this distinction clearer and avoid ambiguity with neuron-to-neuron heterogeneity which refers to differences across cells.
>
> ---
>
> **Q5/C4: Sinusoidal embedding**
>
> Equation (2) is not the embedding at a single time point, but rather a “stack of trigonometric time-series”, since t is a vector of discretised time points of length L. The equation defines a matrix of shape K x L, where K is the number of sinusoids (i.e., frequency components), and each row corresponds to one sine or cosine function evaluated across time. This follows the positional encoding convention introduced in “NeRF: Representing Scenes as Neural Radiance Fields for View Synthesis” by Mildenhall et al. (2020), where the entire set of time samples is embedded using fixed-frequency trigonometric functions. The first Fourier transform of the FNO will act on each of these trigonometric time-series separately, and none of the resulting Fourier transforms are trivial by construction. We clarified the embedding formula in Section 3.3.
>
> ---
>
> **Q6: Clarification on “Cellular Models representing multiple data modalities”**
> Our biophysical PDE neuron models are constructed considering 3 different types/modalities of experimental measurements including the neuron morphology, electrical properties (electrophysiology), and ion channel gene expression (transcriptomics).
>
> ---
>
> **C1-2-3: Clarification on FNO notation and Figure 1**
>
> The operator $R_{\phi}$ is a learnable Fourier multiplier, which can be interpreted as the Fourier transform of a periodic function, consistent with the terminology used in the original FNO paper by Li et al. (2021). Similarly, the expression $\sigma (W + K + b)$ follows the same convention and is explained in detail in our appendix and in the original FNO paper.
>
> The diagram in Fig 1A presents a high-level overview of NOBLE, in which the Neural Operator block internally contains the full architecture of a neural operator. In our study, we use an FNO as described in Li et al. (2021). The parameters referred to in Eq. 1 are all encapsulated within the Neural Operator block in our diagram. Since our focus in Fig 1A was to convey the overall structure of the framework (including the embedding space and conditioning mechanism), we chose to abstract away the internal layers of the neural operator. However, these follow the standard FNO implementation, and a more detailed representation of the FNO can be found in Fig 8.
>
> ---
>
> **C5: Superimposing PDE and NOBLE**
>
> We updated the figure by superimposing the PDE and NOBLE traces, which indeed makes the comparison more direct. Additionally, we have enlarged the figure to improve clarity.

---

> ### Comment · Reviewer_d3WE · 2025-08-05
> **Capturing intrinsic variability with ML approaches**
>
> Thank you for the clarifications.
>
> W1: "Now, other existing ML approaches [28-33;37-40] in neuroscience are not equipped to capture variability" I find difficult to believe that intrinsic variability is not captured by previous ML approaches. For example the early work of Paninski et al. 2004 (Neural Computation 16, 2533–2561) estimated the distribution of the membrane potential (intrinsic variability as described by the authors) of an integrate and fire neuron model. Their approach being based on generalized learn models can extend to more complicated neuron models.

---

> > ### Author Response · Authors · 2025-08-06
> >
> > We thank the reviewer for bringing the work of Paninski et al. (2004) to our attention. It is indeed a foundational and insightful example of a statistical approach to modeling trial-to-trial variability via a probabilistic analogue of a point-neuron model. We agree that such approaches offer ways to introduce stochasticity in a different way and can be extended to more complex neuron models. **We have revised the manuscript to include a discussion of this line of earlier work, along with the relevant references.**
> >
> > We clarify that our original statement was primarily referring to recent deep learning-based approaches for modeling neuron dynamics, which have predominantly focused on deterministic formulations [28-33;37-40]. We also clarify that by intrinsic variability, we meant both to trial-to-trial variability observed within a single neuron and to variability observed across and within different cell types. That said, you are absolutely right that this is not the only class of methods exploring ways to model variability in neuron models. Accordingly, we have revised the statement to explicitly state that it is referring to deep learning-based approaches. In addition, we have identified several other prior works, beyond Paninski et al. (2004), that introduce stochasticity into single-neuron models to capture trial-to-trial variability (e.g., O’Neill et al., 1986; Koyama & Kass, 2008). These studies are now cited and discussed in the revised manuscript.
> >
> > Although Paninski et al.’s approach could, in theory, be extended to more complex systems, limitations in computational hardware at the time restricted their analysis to much simpler dynamics. They were limited to point models with a small number of parameters and could not employ models capable of capturing variability themselves. To compensate, they introduced variability by synthetically injecting white noise, which the authors themselves noted is unrealistic in many situations. We illustrated for instance in Figure 5 how small perturbations can cause large deviations in activity in the nonlinear, high-dimensional PDE-based “Hall of Fame” models, which involve many interdependent conductances and kinetics. Injecting synthetic stochasticity into such models introduces perturbations that can lead to unrealistic predictions, as previously demonstrated for simplistic point-neuron models in (Schneidman et al., 1998; White et al., 2000; Goldwyn & Shea-Brown, 2011).
> >
> > Unlike the above-mentioned prior works that inject synthetic stochasticity to mimic variability, our paper employs an evolutionary optimization strategy to generate a collection of deterministic PDE-based models. These models reflect real conductance-based variability and capture mechanistic differences across cellular states, allowing the ensemble to collectively capture the observed variability in a principled and biologically grounded way. While simply injecting variability may seem sufficient, it does not imply the underlying mechanisms are biologically accurate, as variability in features can arise from many factors, and collapsing them into a stochastic process is a black box simplification. This makes our approach fundamentally more biologically interpretable.
> >
> > The resulting ensemble of PDE-based models serves as the basis for NOBLE, which introduces a set of interpretable feature embeddings that parameterize the PDE model space, allowing for robust sampling and interpolation within a learned latent space of models. At each sampled point, NOBLE can be used to extract electrophysiological features and generate heatmaps that visualize how these features vary across the latent space. This approach not only reveals relationships between latent and observed features but also illuminates the geometry of the learned representations, offering new insights into the underlying structure, variability, and diversity of neuronal behavior.
> >
> > We appreciate the reviewer’s thoughtful comment and updated the manuscript to include these clarifications, along with the relevant references.

---

### Official Review · Reviewer_1sYR · 2025-06-05

**Clarity:** 3
**Significance:** 2
**Originality:** 1
**Rating:** 3
**Confidence:** 4

**Summary:**

This paper introduces NOBLE, a neural operator framework with biologically-informed latent embeddings, to address the challenges of modeling neuronal dynamics. It targets the key problems of high computational cost and the failure of existing models to capture the experimental variability inherent in biological neurons. The authors propose a deep learning solution that learns a direct mapping from an interpretable, continuous space of neuron features to the somatic voltage response following a current injection. Trained on data from biophysically realistic simulations, NOBLE can generate distributions of neural dynamics, effectively accounting for variability. A central contribution is its ability to interpolate within this latent embedding space to create novel, yet biophysically plausible, neuron models. This approach yields a significant 4200x speedup over conventional numerical solvers and is validated against real experimental data, demonstrating its potential to advance the scale and realism of brain circuit modeling.

**Questions:**

The findings presented are compelling, yet several questions remain regarding the framework's broader applicability and robustness. It would be beneficial to understand the methodology for selecting the embedding features. How might one identify a sufficient set of interpretable features for neurons with different properties, such as bursting or adaptation, and how does the model's performance scale with a higher-dimensional embedding? The paper demonstrates excellent performance with interpolation, but its behavior when extrapolating to models outside the training distribution is unknown; clarifying the model's performance degradation in this regime would help define its valid operational domain. From a practical standpoint, the data efficiency of the framework is also a key concern. Understanding how performance varies with the number of initial models used for training would be crucial for assessing the trade-off between the upfront simulation cost and the resulting surrogate's accuracy. Lastly, further discussion on the impact of the chosen time-series subsampling on the ability to capture very fast neuronal dynamics would be valuable.

**Ethical Concerns:**

["NO or VERY MINOR ethics concerns only"]

**Final Justification:**

The starting point of the work is not novel, ignoring a lot of previous work about single neuron embeddings based on electrophysiological activities.

The work is entirely based on simulated data, and the generalization of real scenes needs to be considered.

The exploration of the resulting embeddingsn is not sufficient, and the association at the molecular, structural, and other levels is not fully discussed. Other similar work has relevant content.

I choose to reject it. Thanks!

**Limitations:**

The authors have commendably provided a forthright discussion of the work's limitations within the paper's conclusion. They correctly identify that the framework is currently developed for single neurons and that the biologically-informed embedding space is low-dimensional and constructed manually from pre-selected features. They appropriately position these limitations as clear and exciting directions for future research. Specifically, they suggest that natural extensions would involve scaling the method to multi-neuron networks and exploring the potential of learnable, higher-dimensional embedding spaces. This transparent handling of the current boundaries of the work is a positive aspect of the paper.

**Quality:**

2

**Strengths And Weaknesses:**

The paper is elegantly shown to succeed where direct interpolation of model parameters fails, highlighting its unique value. Furthermore, the technical quality of the study is high, featuring a rigorous validation process on models derived from human neuron data. The framework's performance is not only evaluated on voltage trace accuracy but is also benchmarked against a wide array of electrophysiological features, ensuring its neuroscientific relevance.

Despite these considerable strengths, the work has weaknesses related to its scope and generalization:

The core idea for this framework is similar to NeuPRINT [1]. The authors should compare their framework with NeuPRINT.

The demonstrated embedding space is low-dimensional and hand-engineered for a specific type of neuron, the PVALB interneuron. It remains an open question how this feature selection process would generalize to other neuron types with different dynamic characteristics.

The validation is also confined to models of a single neuron, and demonstrating the framework's effectiveness on a different cell type would have greatly strengthened its claims of general applicability.

While the framework dramatically accelerates inference, it still depends on an initial set of models generated through a computationally expensive process. The method does not eliminate this upfront cost but rather amortizes it, a practical consideration for its adoption.

[1] Learning Time-Invariant Representations for Individual Neurons from Population Dynamics. NeurIPS 2023

---

> ### Author Rebuttal · Authors · 2025-07-31
>
> We thank the reviewer for their thoughtful summary and generous assessment. We are particularly encouraged by their recognition of NOBLE’s ability to generate biophysically plausible dynamics through latent space interpolation, and of its strong validation against both simulation and experimental benchmarks. We also appreciate the reviewer’s acknowledgment of our efforts to go beyond trace accuracy by evaluating a broad set of e-features, which we believe is essential for meaningful neuroscientific application. We have addressed all raised concerns below and hope our response satisfactorily resolves the reviewer’s points. We are happy to provide further elaboration or modifications during the discussion period.
>
> ---
>
> **W1: NeuPRINT and NOBLE are fundamentally different, but complementary**
>
> NeuPRINT is a powerful framework, but deals with a very different dataset. It generates robust representations of optical physiology recordings from labeled neurons in living rodents. While our study also leverages neural data from a labeled cell type (in our case using patch-seq labeling), the dynamics are vastly different. In vivo optical recordings are typically slow (resolution ~ tens of ms) and do not capture single action potentials. The in vitro recordings we train on have sub-ms resolution and capture the full complexity of a neuron’s electrophysiological behavior (e.g. spike inflection point, rise time). Thus, NeuPRINT and NOBLE capture opposite parts of the spectrum: the former focuses on slower fluorescence signals recorded in vivo, the latter on fast intracellular membrane dynamics recorded in vitro.
> Their fundamentally different dynamics make basic comparisons difficult. While Ca-based fluorescence physiology is viable in rodents, these experiments cannot be pursued in humans as they involve genetic manipulation. An alternative would be to develop NOBLE models for rodent neurons to compare the 2 frameworks but, even then, a direct comparison is difficult between the 2 datasets (intracellular patch-clamp voltage recordings with a sampling rate of 20kHz vs. transmembrane fluorescence of Ca-indicators with a sampling rate of >30 Hz). The most direct comparison would be for NeuPRINT to train on human in vitro intracellular physiology data or for NOBLE to train on rodent in vivo optical physiology data.
> In an ideal scenario, NeuPRINT and NOBLE will inform each other. We discuss NeuPRINT in our updated paper and suggest ways for them to interact.
>
> ---
>
> **W2: We demonstrate that the same low-dimensional, biologically interpretable embedding space generalises effectively across neuron types, including VIP neurons**
>
> To demonstrate that NOBLE generalises beyond a single cell type, we trained a new model on a vasoactive intestinal peptide (VIP) neuron using the same embedding construction and e-features. After 450 epochs (using the same architecture and optimizer), NOBLE achieves errors:
> * Relative L2: 2.5%
> * Spikecount: 9.2%
> * AP1 width: 22%
> * Mean AP amplitude: 10%
>
> These results are consistent with those obtained on PVALB neurons, supporting the robustness and generalisability of our embedding strategy. We included this new experiment in the revised paper.
>
> ---
>
>
> **W3/Q4: NOBLE amortises this cost of costly HoF generation and enables fast, scalable synthesis of biorealistic responses**
>
> Training NOBLE on 84,000 samples was completed in ~4 days using a 64GB NVIDIA Tesla P100 GPU (300 epochs). The significant advantage arises post-training: NOBLE enables the instantaneous generation of an infinite number of new HoF model responses within the embedding space's convex hull (Fig. 3). This is a dramatic efficiency gain compared to the 600k CPU hours required to produce the 60 HoF models from which our training data is derived using the traditional evolutionary optimization.
>
> We added an ablation study by varying the number of HoF models used to generate the training data. The analysis below highlights the minimal information NOBLE needs to learn robust underlying representations, allowing for the generation of novel biorealistic HoF responses.
>
> #HoFs excluded|rel L2|Steady State Voltage|Spikecount|AP1 Width|Mean AP Amplitude
> -|-|:-:|:-:|:-:|:-:
> 10|11.7%|2.0%|9.2%|350%|14%
> 20|10.9%|1.8%|19%|920%|17%
> 30|10.9%|1.9%|20%|3004%|20%
> 40|10.6%|1.9%|45%|1698%|20%
>
> Although the relative L2 error on voltage traces remains largely stable, a clear degradation is observed in spike-related features, particularly spike count and AP1 width, as training diversity decreases. This ablation has been added to the paper since it provides an important perspective to the discussion, illustrating that model diversity, not just quantity, is crucial for learning robust representations of neural dynamics.
>
> ---
>
> **Q1/2: Feature selection for embedding**
>
> There is no universally optimal strategy. This requires a supervised decision about which features best capture the behaviour or phenotype of interest. Our goal was to test whether NOBLE could succeed using a minimal yet meaningful embedding space. We thus selected 2 widely used e-features derived from each HoF model’s frequency–current (F–I) curve: the threshold current and the slope at threshold. These features are standard in the neuroscience literature, providing a natural 2D space to represent both firing and non-firing dynamics, and offering strong biological interpretability. We plan to explore more complex or multimodal embeddings in future work.
> More generally, to find features that are both distinct and not redundant, we can quantify the variation of a large set of features for data across/within cell-types as well as feature correlations to determine which subset of features might be suitable for constructing an interpretable embedding space. We note also that within the NOBLE framework the suitability of features (like adaptation and bursting) depends on the level of sophistication of the PDE-based models. NOBLE can account for various complex properties of neuronal membrane dynamics as long as these are represented in the original HoF models NOBLE is trained on.
>
> To quantify the contribution of our feature embeddings, we conducted a comprehensive ablation study systematically varying the number of embedded features. We evaluated five distinct embedding configurations:
> * No embedded features
> * Only $I_{thr}$
> * Only $s_{thr}$
> * $s_{thr}$ and $I_{thr}$
> * $s_{thr}$, $I_{thr}$ and AHP depth
>
> AHP depth was chosen as it showed large variations across samples and low correlation to sthr and Ith. The results of this ablation study, detailing the quantitative effects on NOBLE's predictive performance, are presented below.
>
> Features embedded|rel L2|Steady State Voltage|Spikecount|AP1 Width|Mean AP Amplitude
> -|-|:-:|:-:|:-:|:-:
> None|12.1%|1.31% |Never fires|Never fires|Never fires
> $s_{thr}$|2.83% |1.33%|4.9%|233%|13%
>  $I_{thr}$|2.73%|1.20% |4.4%|107%|14%
> $s_{thr}$ and $I_{thr}$|1.92%|1.02%|3.1%|27%|8.9%
> $s_{thr}$, $I_{thr}$ and AHP depth|2.16%|1.04%|3.3%|22%|9.5%
>
> These results show that without the use of embeddings, NOBLE is unable to predict any firing response successfully. Embedding either slope or intercept enables comparable L2 accuracy, with intercept yielding better AP1 width prediction. The best overall performance is achieved by embedding both, with slight additional gains from including AHP depth. We agree that such an ablation adds value to the use of embeddings and has been added to the paper.
>
> ---
>
> **Q3: Extrapolating to models outside the training distribution**
>
> We agree with the reviewer that NOBLE “demonstrates excellent performance with interpolation” and realize that the paper may not sufficiently highlight the aspect of “extrapolating to models outside the training distribution”. The rightmost plots in Fig. 6 directly addresses this concern, demonstrating NOBLE's robust generalization to unseen, out-of-distribution models. Specifically, these plots show the predicted voltage responses, for different input currents, with 200 randomly sampled unseen HoF models from the convex hull of the embedding space (see Fig 3).
>
> ---
>
> **Q5: Subsampling strategy and fast neuronal dynamics**
> Our subsampling rate was carefully chosen so it can resolve the highest frequency neuronal dynamics (such as action potential halfwidths (~0.05ms)) in response to input currents in the considered ranges, for the PVALB neuron selected. As detailed in Section 3.1 and Figs 9-10, we evaluated multiple subsampling strategies and quantified their effects on e-features accuracy. This analysis allowed us to identify a conservative upper bound beyond which subsampling would degrade signal and e-features fidelity. The chosen subsampling factor balances computational efficiency with the need to capture very fast neuronal dynamics

---

> > ### Comment · Reviewer_1sYR · 2025-08-01
> >
> > Dear Authors,
> > Thank you for your feedback.
> > I still do not consider NOBLE to be entirely novel.
> > I maintain the view that NOBLE lacks proper comparisons with relevant works such as NeuPRINT[1] and NEMO[2] in an appropriate manner.
> >
> > [1]Learning Time-Invariant Representations for Individual Neurons from Population Dynamics. NeurIPS 2023
> >
> > [2]In vivo cell-type and brain region classification via multimodal contrastive learning ICLR 2025

---

> > > ### Author Response · Authors · 2025-08-03
> > >
> > > We thank the reviewer for the prompt response and motivating a deeper discussion with other, more established, neuronal embedding methods. We revised the paper to include the discussion below
> > >
> > > The following table highlights their key distinctions
> > >
> > > ||NOBLE|NeuPRINT|NEMO
> > > |-|-|-|-|
> > > |Core Problem|Modeling intracellular dynamics| Population context modeling and learning discriminative representations|Learning discriminative embeddings
> > > |Embedding Purpose|Generation of new biorealistic models|Downstream classification|Classification and clustering of cell type and brain region
> > > |Input|Injected current|Neuron and population activity history|Spike waveform and autocorrelogram
> > > |Output|Somatic voltage response|Latent embedding for classification|Cell type and brain region labels
> > >
> > > We now clarify their distinct goals and highlight the specific novelty of NOBLE
> > >
> > > **NeuPRINT**
> > >
> > > While NeuPRINT is a valuable contribution to learning neural representations, its scientific goals and applications are distinct to NOBLE
> > > * **Causal vs. Contextual Modeling.** NOBLE addresses causal surrogate modelling while NeuPRINT addresses population context modeling. NOBLE learns a direct, causal mapping from a controlled input to a single neuron’s voltage response. It serves as a fast, accurate surrogate of biophysical simulators to study how a neuron's intrinsic properties shape its response to an intracellular current. This type of modeling enables designing current input waveforms tailored to interrogate and measure the response from individual conductances, relevant under in vitro conditions where the input to a cell can be controlled. NeuPRINT, in contrast, does not model a direct input-output relationship, but a neuron's activity as a function of its own history and the collective ongoing dynamics of the surrounding population. This is relevant under in vivo conditions where the activity of multiple neurons can be monitored, but it is impossible to isolate the intracellular input to any single neuron. The goal is to understand how population activity influences a neuron's activity, rather than to simulate the neuron’s response to a specific isolated input
> > > * **Generative vs. Discriminative Embeddings.** The purpose and utility of the learned latent embeddings are distinct. The primary innovation of NOBLE's latent space is its generative capability. We construct an interpretable, biologically-informed continuous embedding allowing the generation of a potentially infinite number of novel biorealistic neuron models (intractable with the original PDEs) reflecting experimental features and their variation. The goal of NeuPRINT's learned representation is discriminative. It learns a time-invariant embedding to serve as a feature vector in a downstream classification task. The embedding's value is measured by its classification accuracy, not its ability to generate new functional models
> > > * **Fast Intracellular vs. Slow Population-Level Dynamics.** The two frameworks capture phenomena at opposite ends of the electrophysiology spectrum. NOBLE is designed to capture the fast dynamics of a single neuron's intracellular voltage. It is trained on data from detailed biophysical PDE models validated against in vitro intracellular whole-cell patch-clamp recordings. This allows NOBLE to resolve sub-ms features of individual action potentials, such as spike initiation, afterhyperpolarization, and other complex nonlinearities. NeuPRINT operates on slower in vivo 2-photon calcium imaging data, which has a temporal resolution on the order of tens of ms. This modality measures a slow-integrating proxy of neural activity (calcium fluorescence) and does not capture individual action potentials
> > >
> > > **NEMO**
> > >
> > > While NOBLE and NEMO operate on electrophysiology data, they address fundamentally different scientific questions
> > > * NOBLE is a neural operator-based framework designed as a fast, generative surrogate for single-neuron intracellular dynamics, predicting voltage dynamics from a given input
> > > * NEMO is a self-supervised, multimodal contrastive learning framework that learns discriminative embeddings for cell-type and brain-region classification. It jointly embeds a neuron's spike waveform (using an MLP) and its activity autocorrelogram (using a CNN). These embeddings are mapped via a learnable linear projector into a final projection space where a contrastive loss is optimized. NEMO is trained to produce a well-structured representation space suitable for downstream classification and clustering tasks
> > >
> > > Despite these differences, NOBLE can complement NEMO. NEMO requires large-scale electrophysiology recordings for training. Given NOBLE's ability to generate novel bio-realistic neuron models, it could serve as a data augmentation engine for NEMO by producing synthetic yet plausible electrophysiology, characteristic of a specific cell (but clearly not limited to that), leading to larger, more diverse training sets, enhancing the robustness and performance of NEMO

---

> > > > ### Comment · Reviewer_1sYR · 2025-08-03
> > > >
> > > > Dear Authors,
> > > >
> > > > I think NeuPRINT can also achieve the functionality of NOBLE. NeuPRINT can easily be modified to accept injected current as input. NeuPRINT takes a more advanced approach to modeling the inputs received by a single neuron in vivo, namely inputs from surrounding neurons and external inputs. I think you should conduct concrete experiments to compare their architectures, rather than conceptual comparisons.
> > > >
> > > > NeuPRINT is also a temporal causal modeling approach based on the transformer with masked attention.
> > > >
> > > > [1]Learning Time-Invariant Representations for Individual Neurons from Population Dynamics. NeurIPS 2023
> > > >
> > > > -- Reviewer 1sYR

---

> > > > > ### Comment · Reviewer_1sYR · 2025-08-03
> > > > >
> > > > > Supplement: If you are modeling using a causal framework, you should be careful to remove some biases and make counterfactual estimates, rather than simply modeling based on the observed data in a data-driven manner.
> > > > >
> > > > > -- Reviewer 1sYR

---

> > > > > > ### Author Response · Authors · 2025-08-04
> > > > > >
> > > > > > ## Causality
> > > > > > We thank the reviewer for raising this point and appreciate the opportunity to clarify. We apologize if our previous response caused any confusion when referring to a “causal mapping from a controlled input to a single neuron’s voltage response”. By “causality,” we intended to convey the causal effect of a current injection on a single neuron's dynamics, meaning the prediction of the voltage response does not depend on the broader network or population voltage history. In this scenario, the only relevant input is the intracellular injection and its characteristic. We did not mean to suggest causal inference in the formal statistical sense.

---

> > > > > ### Author Response · Authors · 2025-08-04
> > > > >
> > > > > We thank the reviewer for the opportunity to further clarify the contributions of our work.
> > > > >
> > > > > ## Suitability of the Suggested NeuPRINT as a Benchmark
> > > > >
> > > > > As detailed in our previous response, the two frameworks differ fundamentally in three key aspects: their scientific goals (causal vs. contextual modeling), their core function (generative vs. discriminative embeddings), and their underlying data modality (fast intracellular vs. slow population-level dynamics). Given these core differences in purpose, function, and data, NeuPRINT would need to be significantly modified to allow for a direct comparison. Several alternative architectures are both more promising and more appropriately aligned with the requirements of meaningful benchmarking on this task than NeuPRINT, and although none can be used out-of-the-box, they would require far less restructuring than NeuPRINT would. The main relevant reusable component in NeuPRINT is the transformer architecture, which is itself not an original contribution of NeuPRINT. Therefore, **NeuPRINT is not a suitable benchmark for evaluating our method**.
> > > > >
> > > > > ## Availability of Relevant Benchmarks
> > > > >
> > > > > A meaningful comparison would be against other methods modeling the membrane voltage dynamics of single neurons in response to direct somatic current injections, such as the ones cited in our literature review. However, while they are relevant to the general problem, they do not address the core innovations of our framework. Our paper presents the first model demonstrated to capture both the full range of the electrophysiological spectrum and, critically, to exhibit realistic trial-to-trial variability. This is a key feature of biological neurons absent in prior deterministic models. This means **there are currently no established benchmarks available for evaluating our model's unique capabilities**. Developing such benchmarks would require novel research to adapt existing methods to our setting, which we consider beyond the scope of this paper.
> > > > >
> > > > > ## Rationale for Our Evaluation Strategy
> > > > >
> > > > > Given the lack of direct benchmarks, one could consider creating a new comparison by adapting another state-of-the-art model, such as a different neural operator or a sequence-to-sequence transformer. However, we argue that such a comparison would not add significant scientific value. Our model already performs well in capturing the dynamics of the underlying ensemble of PDE simulation data. The PDE solvers used to generate the dataset are themselves approximations of biological reality with non-negligible errors.. As a result, **further reducing prediction errors risks overfitting to the PDE simulations without substantially better capturing the dynamics and e-features of the true real-life recordings**. The more significant scientific challenge we address is building a generative framework that captures the complex and variable voltage responses of single neurons to controlled intracellular inputs, a task previous deep learning approaches have not successfully achieved. Thus, our evaluation is deliberately focused on validating these novel capabilities. We demonstrate that our model generates outputs that reproduce key statistical and biophysical features observed in experimental recordings, proving its utility as a new type of scientific tool. We believe this focus on a new modeling paradigm is a more significant and impactful contribution than chasing (potentially insignificant) incremental gains on a predictive metric.
> > > > >
> > > > > ## Innovation
> > > > > Given the absence of prior work with comparable methods or applications, we find it difficult to understand the basis for the lowest originality score of 1. Based on the latest responses, it appears that the reviewer suggests NOBLE lacks originality unless a direct comparison is made with NeuPRINT, a method developed for a fundamentally different context, core problem, data set, and data type. Importantly, the reviewer does not appear to question the novelty of our approach relative to other neural operators, physics-informed neural networks (PINNs), or broader machine learning techniques. Furthermore, from the reviewer’s comments, there seems to be no concern regarding the quality or informativeness of the embeddings generated by NOBLE, and the novelty in that aspect remains uncontested. In light of the clear novelty and distinctive contributions of our work, we respectfully ask the reviewer to reconsider the originality score. Alternatively, the reviewer can clarify the specific concerns that led to such a low assessment, especially given that NeuPRINT is not an appropriate benchmark as it differs fundamentally from our approach in context, data type, and objectives, and thus should not be used to question the originality of our contribution.

---

### Official Review · Reviewer_VJJQ · 2025-06-25

**Clarity:** 3
**Significance:** 3
**Originality:** 3
**Rating:** 4
**Confidence:** 4

**Summary:**

The manuscript introduces Neural Operator with biologically-informed Latent Embeddings (NOBLE). In this framework a neural operator, or more specifically a Fourier Neural Operator (FNO) is trained to predict the somatic voltage from a f(I) curves for an entire collection of biophysical (Hall of Fame aka. HoF) models. This yields a single representative model for a diverse set of HoF models which collectively represent the dynamics of a single cell. This NOBLE model can then be used to generate biophysically plausible responses at any point within the embedding space, while being much cheaper to evaluate than the original biophysical models.

**Questions:**

- Does the analysis done in Fig. 6 also work for the test data? It would be nice to see this hold there.
- In L274 you mention a "relative L4 error" and in L290 you say "L2 error". If this is not a typo, could the authors explain why they picked 2 different metrics and add a formula for the rel. L4 error to the appendix? Was the error computed on the full or downsampled traces?
- I am not very familiar with FNOs. Could the authors comment on whether sampling time is affected by the fidelity of the samples? If yes, does the 4200x speedup refer to the downsampled version of the voltage trace?
- In Fig 7. how does NOBLE compare on the remaining summary statistics? This would be great to have in the appendix.
-

**Ethical Concerns:**

["NO or VERY MINOR ethics concerns only"]

**Final Justification:**

The authors have done extensive revisions of the manuscript and addressed all my major concerns during the rebuttal. While the novelty beyond the use of FNOs is limited in my opinion (training of a neural emulator + embedding space), I think the work should be published. I hence err on the site of acceptance.

**Limitations:**

I feel like technical limitations were insufficiently discussed. For example how does NOBLE compare when the training budget is factored in? What are limitations of FNOs that also apply to NOBLE?

**Paper Formatting Concerns:**

- L276 "generate**s**"
- L288 Might mean Fig. 4**B**
- Fig 5. caption experimental **data**

**Quality:**

2

**Strengths And Weaknesses:**

## Strengths
- Using FNOs to capture the variability across a collection of biophysical models in a single emulator by encoding this into a shared latent space is a neat idea. With a more interesting latent space, i.e. including more behavioral features or multimodal features one could imagine training a "foundation model" for a certain cell type this way.
- Another advantage of FNOs is the ability to generate dynamics at higher resolution than what was trained on, which can be especially helpful for generating voltage traces in the hundreds of milliseconds. (This was unfortunately not shown in the paper though).

## Weaknesses
- The main argument the paper seems to be making, is that the NOBLE surrogate is a _better_ and _more flexible_ model of the data than a family of HoF models, however I feel like that this was not demonstrated sufficiently. While the authors show that NOBLE approximates the HoF models well in different settings, the authors comparison of NOBLE predictions with the original data is limited (only Fig. 7 really does this for a subset of features). To show that the NOBLE surrogate is indeed a potentially _better_ model of the data it would be great to find a latent input (or set of inputs) that match the data better than the HoF models (or come up with another metric which shows NOBLE models are superior to HoF models). In addition the 2nd selling point of a NOBLE model, _more flexibility_, seems underutilized to me. More examples of how the latent space can be used would effectively, i.e. by showing the distribution of summary statistics across the entire latent space  (i.e. via pairplots) and comparing them to the data. An interesting comparison would also be to see the NOBLE predictions in a small ball around the f(I) of the data / for a HoF model.
- In the introduction the authors allude to patch-seq datasets and how one can use them to study how genes might relate to circuit or brain function, but this is not picked up anywhere in the discussion about NOBLE (or I have missed it). If you could point me to this in the text or elaborate more on how NOBLE can help with this, that would be appreciated.
- The paper would benefit from more polished figures. The line thickness in Fig 2. is really small (I printed the paper out) and it would be nice to include scale bars or axes. The side by side arrangement of Fig 3. makes it hard to compare HoF and NOBLE predictions. Overlaying them would make this more obvious. Also the in and out of distribution currents appear to be the same. It would also be great to show  distributions of important summary features rather than the full voltages traces in Fig. 6, as it is hard to judge how the NOBLE models compare here.

## Comments
- In Fig 7. I'd argue that report medians and quantiles makes more sense. Reporting the $\mu-\sigma$ suggests existance of negative spike counts.
- In Fig 5. it would be nice to also see NOBLE compared to the experimental data. In addition the plot could be improved by replacing the lines with error bars or a shaded area. Also why did you use seemingly different models in A and B? This makes it a bit hard to compare.
- The PDE-based biophysical models are in fact ODE-based. While multicompartment biophysical models could be seen as spatially discretized PDEs, what NEURON simulates is a system of coupled ODEs, **not** PDEs.

---

> ### Author Rebuttal · Authors · 2025-07-31
>
> We thank the reviewer for their insightful comments and their recognition of NOBLE’s core contribution: learning a unified neural operator capturing variability across biophysical models. We also appreciate their perspective on foundation models. We have addressed all raised concerns below and hope our response satisfactorily resolves the reviewer’s points. We are happy to provide further elaboration or modifications during the discussion period.
>
> ---
>
> **NOBLE’s purpose (W1/Q1)**
>
> The goal of NOBLE is not to outperform individual HoF models in matching experimental data. Each HoF model is already optimized to closely fit experimental recordings. Any instance where NOBLE achieves a closer fit would be accidental. Instead, NOBLE is designed to generalize across a biologically realistic population of neuron models, interpolate within a latent feature space to generate new biorealistic models, and dramatically reduce computational costs
>
>
> The rightmost plots in Fig 6 illustrate NOBLE’s predictive performance on held-out HoF models from the test set, by showing predicted voltage responses at different input currents for 200 randomly sampled models. These models were not seen during training, yet NOBLE's predictions closely match the solver outputs. This strong alignment across a wide set of out-of-distribution models highlights NOBLE’s ability to generate biorealistic responses even in previously unseen regions of the latent space. Also note that during testing, NOBLE is evaluated on random input current values that were not experienced during training
>
> ---
>
> **Interpretability (W1/W2)**
>
> We appreciate the reviewer’s insight and are glad they anticipate directions we are pursuing. One promising approach involves sampling the embedding space, running NOBLE on each point, and computing e-features to generate heatmaps. This allows to visualise how features vary across the latent space (thereby informing about the relation between the features visualized and those defining the latent space), helps highlight the geometry of the learned representations while also offering new insights into the underlying structure, variability, and diversity of neuronal behaviour. We added an example of such a heatmap and highlighted this as an exciting direction to explore further in future work
>
> Regarding the use of patch-seq data, the 4 inhibitory cell-types presented in Fig 2B exhibit distinct gene expression profiles and ion channel expression. By training a large-scale version of NOBLE that captures neurons from multiple cell-types with patch-seq data included in the embedding space, one then has an efficient tool to probe the effect of ion channel expression on neuronal responses across different cell-types. For example, one can embed and perturb the ionic conductances or gene expression concentrations and analyze how these variables affect a neuron’s electrical properties (Fig 5 illustrates how parameters in the embedding space can be perturbed via interpolation using NOBLE, but not with PDE models)
>
> ---
>
> **Subsampling (Q2/Q3)**
>
> The entire numerical experiments are conducted on downsampled voltage traces. Our downsampling analysis confirmed that the chosen resolution preserves all relevant e-features (see Section 3.1). In particular, NOBLE is trained on the downsampled traces. The voltage and e-feature errors and the computational times are also computed on the downsampled traces. Now, NOBLE can be queried and remains consistent at higher resolutions (it essentially spectrally interpolates). We tested inference at the original resolution and observed no noticeable difference in computational time compared to predictions at the 3x downsampled resolution.
>
> ---
>
> **Training loss (Q2)**
>
> The use of L4 for training is motivated by an overfitting study conducted before training on the full dataset where we trained the FNO on a small number of samples using different loss functions to assess their ability to fit voltage traces and preserve key e-features. L4 consistently preserved spike-related features more effectively than L2, particularly in capturing spike amplitudes and widths. Based on these results, we selected L4 as the training loss. We added the details of this overfitting study and the formula for the relative Lp error to the appendix. We only use the relative L2 error for reporting as it is a more standard metric.
>
> ---
>
> **“PDE-based” Terminology  (C3)**
>
> While the neuron models are based on PDE dynamics (specifically the cable equation), their simulation in NEURON relies on a spatial discretization that transforms the system into coupled ODEs. We use the term “PDE-based biophysical models” to refer to models that follow the mathematical formulation of the cable equation. We added sentences to clarify this terminology in the revised paper
>
> ---
>
> **Local sampling around HoF model (W1)**
>
> This analysis is addressed in Figs 4B and 5B. Fig 4B shows time series responses for a HoF model excluded from training. Fig 5B presents the F-I curves for 50 NOBLE samples drawn from a small ball around the excluded HoF model, along with the voltage trace of one representative sample. A new Appendix figure shows the time series for all 50 sampled models, complementing Fig 6. This confirms that the predicted dynamics remain close to the behaviour of the original HoF model, further highlighting NOBLE’s local continuity in the embedding space
>
> ---
>
> **Figure-based modifications (W3/Q4/C1/C2)**
>
> * Fig 2: We increased line thickness to enhance readability
> * Fig 3: We agree that superimposing the traces improves interpretability and revised the paper accordingly
> * Fig 3,4,5,6: We clarified the captions. If interested, the revised versions can be found in the "Figures and Captions" section of our response to Reviewer 5ihX
> * Fig 5: The same models were used in both panels, but with different stimulus durations: 1s in 5A (to match experimental data) and 0.4s in 5B (used for NOBLE). The firing rate is not uniform over time, which explains the differences observed in the F–I curves. We do not compare time series between NOBLE and experimental data directly, due to the mismatch in stimulus duration. Instead, we compare summary features in Fig 7, as these are mostly not affected by stimulus length. For spikecount, we approximate a 1s value by linearly scaling the 0.4s response, although we recognise this is only an approximation.
> * Fig 7: We agree that reporting mean and std could misleadingly suggest the possibility of negative spike counts. We now report the median and interquartile range which more appropriately reflect the distribution of e-features. Fig 7 is used for ensemble performance visualisation and shows distributions of 4 key e-features compared to experimental data. This complements the feature-level analysis by illustrating the biophysical realism of the generated time series. Following the reviewer’s suggestion, we added a full comparison of NOBLE’s predictions to experimental recordings across the remaining e-features in the appendix
>
> ---
>
> **1. Limitations**
>
> We extended the discussion relating to the limitations of NOBLE, and created a new section dedicated to that discussion
>
> **Training Budget (L1)** This is a great question. While training NOBLE involves an upfront cost, it offers significant long-term efficiency and flexibility compared to traditional modelling approaches. Once trained, NOBLE enables the unlimited generation of new, biophysically realistic responses through interpolation (not possible with the original PDE models), and fast inference of voltage traces for arbitrary models in the embedding space
>
> Training NOBLE on 84,000 samples for 300 epochs took ~4 days on a 64 GB NVIDIA Tesla P100 GPU. This is small compared to the 600,000 CPU hours required to generate each HoF model via evolutionary optimization. We added this information to the appendix. Note that we trained until the optimization fully converges, but could have stopped earlier and trained on a smaller number of samples while retaining satisfactory performance
>
> We added an ablation study where we varied the number of HoF models used to generate the training data while keeping the number of training samples fixed, and evaluated performance on the original HoF test set
>
> #HoFs excluded|rel L2|Steady State Voltage|Spikecount|AP1 Width|Mean AP Amplitude
> -|-|:-:|:-:|:-:|:-:
> 10|11.7%|2.0%|9.2%|350%|14%
> 20|10.9%|1.8%|19%|920%|17%
> 30|10.9%|1.9%|20%|3004%|20%
> 40|10.6%|1.9%|45%|1698%|20%
>
> Although the relative L2 error on voltage traces remains largely stable, a clear degradation is observed in spike-related features, particularly spike count and AP1 width, as training diversity decreases. We added this ablation to the paper to illustrate that model diversity, not just quantity, is crucial for learning robust representations of neural dynamics
>
> **2. FNO Limitations (L2)**
> We first emphasize that the NOBLE framework is general and not restricted to the use of the FNO as the neural operator
> A general limitation of data-driven approaches is that they inherently reflect the assumptions built into the underlying PDE model and numerical solver. Our ensemble approach helps mitigate this limitation by combining predictions from multiple PDE models, each grounded in different assumptions
>
> We selected the FNO as the most suitable neural operator for our setting, given its strong performance and computational efficiency. It offers key advantages over standard neural networks and is faster than many alternative neural operators, by leveraging FFTs to accelerate internal computations. While the FNO requires inputs and outputs on equidistant grids, this aligns perfectly with our data, as both experimental recordings and PDE simulations are sampled at constant timesteps. We added a justification for this choice in Section 4 and discussed alternative neural operators that may be better suited for scenarios where uniform time sampling is not available

---

> > ### Author Response · Authors · 2025-08-06
> >
> > Dear Reviewer,
> >
> > We hope our response addressed your questions clearly. Please let us know if you have any remaining concerns or if there is anything we can further elaborate on.
> >
> > If our clarifications and revisions to the manuscript have resolved your concerns, we would greatly appreciate if you would consider re-evaluating your score.
> >
> > Thank you again for your time and thoughtful review.

---

> ### Comment · Reviewer_VJJQ · 2025-08-07
> **response to the author's rebuttal**
>
> First of all I beg the authors pardon for the late response and to keep them waiting. I thank them for their thorough rebuttal and for the detailed answers to my questions. Let me try to respond point by point:
>
> - NOBLE’s purpose: As I understand it NOBLE tries to better and more flexibly capture **biological variability**, compared to a set of HoF models. When I referred to _data_, what I meant is to demonstrate that the NOBLE model correctly captures the biological variability (in summary feature space of say PVALB neurons) and does so in a continuous fashion. I feel that this point is not sufficiently shown. In my view Fig. 7 shows this only for a set of 5 inputs and 4 summary statistics, while Fig 6. just shows that NOBLE does not overfit to the training set. The authors could for example show how the summary stats of the NOBLE surrogate vary for a continuous interpolation of the latent space and how that compares to the data. Similar to Fig. 5, but in feature space. This could be added to Fig. 5 for example.
>
> - Interpretability: I appreciate the authors additional inclusion of feature distributions across the latent space. I think this is a valuable analysis that I'd be curious to see the results of.
>
> - Subsampling:
> >We tested inference at the original resolution and observed no noticeable difference in computational time compared to predictions at the 3x downsampled resolution.
>
> The authors could mention this in their manuscript as it is a potential advantage of the NOBLE framework.
>
> - Training Loss:
> > L4 consistently preserved spike-related features more effectively than L2, particularly in capturing spike amplitudes and widths.
>
> I did not know this! This is a super interesting and useful nugget of information that I would love the author's included in their revised manuscript.
>
> - PDE-based terminology:
> > We use the term “PDE-based biophysical models” to refer to models that follow the mathematical formulation of the cable equation.
>
> While biophysical models indeed derive from the cable equation (PDE), the actual model being solved is in fact an ODE. I would argue most practitioners are probably more familiar with the term ODE in this context. I can see that this might be a question of taste though. I appreciate that the authors added a clarifying statement about this to the text.
>
> - Local sampling around HoF model: Thank you for adding the additional analysis. I still feel that showing features would be more intuitive than showing the traces though (also the NOBLE's purpose above).
>
> - Figure based modifications: Why not use spike rate as a feature? This is independent of duration. Thank you for adding the other features to the appendix!
>
> - Training budget: I appreciate the additional ablations! The insights here could be important for considering when a potential practitioner should choose NOBLE over lets say just a few HoF models.
>
> - Limitations: I appreciate that the authors added an additional section to discuss limitations and to justify their choice to use FNOs.
>
> I thank the authors again for their extensive revision of the paper in light of my criticisms. The authors have addressed most of my concerns in their rebuttal. I am in general willing to raise my score, if the authors could respond to the following two points:
>
> Firstly, in my review I raised the point that:
> >In the introduction the authors allude to patch-seq datasets and how one can use them to study how genes might relate to circuit or brain function, but this is not picked up anywhere in the discussion about NOBLE (or I have missed it). If you could point me to this in the text or elaborate more on how NOBLE can help with this, that would be appreciated.
>
> While the authors mention in their rebuttal that this is illustrated by Fig. 2B and Fig. 5, I maintain that it is not sufficiently discussed (or at least not clearly enough in my opinion), despite being a motivation for NOBLE in the first place. If the authors could make this more obvious / add this to the discussion of the results, I think this would be helpful.
>
> Secondly, if the authors could take another shot at responding to my concerns about _NOBLE's purpose_ (see above), I would greatly appreciate this.
>
> Thank you to the authors for replying on such short notice.

---

> > ### Author Response · Authors · 2025-08-08
> >
> > We thank the reviewer for their thoughtful and detailed engagement with both our manuscript and rebuttal. We appreciate their positive remarks on the additions and clarifications we have made, and are encouraged by their willingness to raise their score. Below, we address the remaining concerns regarding the purpose of the NOBLE framework, its connection to patch-seq datasets, and the spike rate y-axis in Fig. 5.
> >
> > ---
> > ### Discrepancy in Fig. 5 y-axis ###
> >
> > In Fig. 5A, we compare experimental data with the Hall of Fame (HoF) models using an activation duration of 1s, whereas in Fig. 5B, the HoF models are compared with NOBLE using an activation duration of 0.4s. The y-axis in both cases represents spike (firing) rate. However, spike rate is a nonlinear function of time. In our initial analysis for Fig. 5B, we assumed a linear relationship and scaled the number of spikes for each amplitude proportionally as if the activation duration were 1s. We recognise that placing the two plots side-by-side with this inconsistency can be confusing. To address this, we have revised the analysis so that the experimental and PDE data in Fig. 5A are trimmed to match an activation duration of 0.4s, ensuring a fairer and more direct comparison of the frequency-current relationship across the experimental data, HoF models, and NOBLE.
> >
> > ---
> > ### Studying How Genes Relate to Brain Function ###
> >
> > We have expanded the discussion on how NOBLE can be applied to study the relationship between gene expression, neuronal electrical properties, and broader brain circuit function. While our paper demonstrates embeddings using only two electrophysiological features (e-features) ($I_{thr}$ and $s_{thr}$ from the frequency-current curve), the framework can readily incorporate additional e-features as well as gene expression data from patch-seq.
> >
> > Recent patch-seq datasets have shown that transcriptomically defined cell types exhibit distinct electrophysiological and morphological properties (Berg et al., 2021; Lee et al., 2023; Gabitto et al., 2024). Yet, the complex mapping between gene expression, electrophysiology, and morphology remains poorly understood, and it is experimentally challenging to probe how specific changes in gene expression affect neuronal electrical properties in the human brain. NOBLE can address this by learning joint embeddings of electrophysiology and transcriptomics, enabling a unified latent space that links these modalities. By interpolating within this space (analogous to Fig. 5), NOBLE can simulate how varying gene expression levels or ionic conductances influence neuronal firing behaviour, providing a computational means to test hypotheses that would be infeasible experimentally.
> >
> > Once trained on patch-seq datasets, this approach could also be used to study the effects of genes associated with neurological diseases (Buchin et al., 2022; Gabitto et al., 2024) and to predict how such changes impact neuronal electrical behaviour. For example, NOBLE could help pinpoint cell types that are particularly sensitive or vulnerable to pathological effects, offering mechanistic explanations for why some populations, such as SST neurons, are disproportionately affected in certain neurodegenerative diseases (Gabitto et al., 2024).
> >
> > ---
> > ### NOBLE’s Purpose ###
> >
> > Our claim is not that NOBLE captures biological variability better than HoF models, but that it does so more flexibly. Its continuous latent space enables the generation of arbitrary biorealistic neuron models for a given cell type without further optimization. In contrast, HoF models are restricted to the finite set produced by evolutionary optimization, and creating additional models incurs substantial computational cost.
> >
> > This flexibility relates directly to our previous response on “Interpretability”. NOBLE’s continuous latent space can be densely sampled to explore how e-features of interest vary across it and their agreement with experimental data. Following the reviewer’s suggestion, we extend Fig. 5/7 and illustrate how the NOBLE surrogate’s summary statistics evolve under continuous interpolation.
> >
> > We detail a potential procedure below:
> > 1. Densely sample models in the latent space
> > 2. Evaluate NOBLE on each corresponding synthetic neuron model for a given input stimulus
> > 3. Compute the e-feature of interest for the sampled models, and calculate its relative error with respect to the corresponding experimental feature
> > 4. Visualize these values as a heatmap overlaid on the latent space
> >
> > Such heatmaps highlight the geometry of the learned representations and provide insight into the underlying structure, variability, and diversity of neuronal behavior. We added such an example of a heatmap using NOBLE. While a similar procedure could in theory be applied to HoF models, Fig. 5A already shows that they do not interpolate well, making the analysis erroneous.

---

> > > ### Comment · Reviewer_VJJQ · 2025-08-08
> > >
> > > Thank you for the prompt response and the additions to the manuscript. While I am still somewhat skeptical about the overall practical usefulness and novelty of the proposed method, I don't have major remaining concerns. I thank the authors for their extensive revision and answering of my questions. I am therefore raising my score.

---

### Official Review · Reviewer_5ihX · 2025-06-27

**Clarity:** 3
**Significance:** 3
**Originality:** 3
**Rating:** 5
**Confidence:** 3

**Summary:**

This contribution proposes a neural operator framework for predicting voltage dynamics of single neurons in response to the input current. The model is a deep network and operates on the latent space of user-defined neuron properties. The latent space is constructed using embedding of observed neuron properties. Compared to previous approaches, the proposed framework is significantly faster, captures better across-trial variability and generalizes better to unseen levels of stimuli.

**Questions:**

1) How is the embedding space constructed?

2) Have the authors tested the performance of the model using current injections that vary over time?

3) Why are Pyramidal (Excitatory) neurons not shown?

4) All tested neurons seem to respond with high firing rates and relatively regular spike trains. Have authors tried to reproduces neural activity with lower firing rates? This might be more challenging because of increased variability of responses in such firing regimes.

5) What is the utility of the Fourier Neural operator?

**Ethical Concerns:**

["NO or VERY MINOR ethics concerns only"]

**Final Justification:**

The paper provides a useful, generally applicable and efficient method to study properties of single neurons from neural data. Also, the method can be extended in various ways, e.g., different objective function, different features, etc.

**Limitations:**

Limitations are briefly discussed in the section “Conclusion”. I suggest to give more emphasis on discussing limitations and to give such discussion a separate section instead of merging it with Conclusion. I suggest to add to limitations the fact that the current contribution does not test the model on weak and time-varying input currents.

**Paper Formatting Concerns:**

No formatting concerns.

**Quality:**

3

**Strengths And Weaknesses:**

_Strengths_:
Submission is technically sound and proposes an important upgrade to previous approaches on reconstructing activity of single neurons. It also outperforms previous approaches on the ability to capture variability across trials, its generalization capacity to reproduce neural responses to current injections that were not sued during training and strongly outperforms previous approaches in speed of learning. Overall, this paper reads as a useful tool to study neural responses of different neuron types.

_Weaknesses_:
1.) No clear metrics of performance is used, even though this would be useful to quantitatively compare the performance of NOBLE with HoF models without embeddings.
2) Along the paper and in particular in the Introduction, a number of acronyms is used, to an extent where this decreases the clarity of the presentation. I advise the authors to limit the number of acronyms to only a handful  (3-4) that are the most well known (besides the acronym of their model). This will ease the reading for readers that are less familiar with the topic.
3) The writing is somewhat convoluted all along the paper. To make the paper more accessible, I kindly suggest to the authors to revise the text so that it gains in clarity. For example, on several occasions the text is unspecific, which decreases clarity. In line 96, authors mention that the latent space is constructed from “observable neuron characteristics”. Later on (line 257) we learn model features that define the latent space are only two, the current at spiking threshold and its slope. It would be clearer to simply state what those features are when they are first mentioned.
4) Several figure captions are too short and do not explain what is shown on the figure. For example, it remains ambiguous what is shown on Figure 3. Figure 4 and 6 show two types of voltages, one seems to be with spikes, the other without spikes. This is not clearly explained in the text of the figure caption.

---

> ### Author Rebuttal · Authors · 2025-07-31
>
> We thank the reviewer for the thoughtful and positive evaluation of our work. We appreciate the recognition of the model’s technical soundness, its ability to capture across-trial variability, its generalization to unseen stimuli, and its substantial improvements in computational efficiency. We are pleased that the reviewer regards our approach as a valuable tool for studying neural responses across neuron types. We have addressed all raised concerns below and hope our response satisfactorily resolves the reviewer’s points. We are happy to provide further elaboration or modifications during the discussion period.
>
> ---
>
> ## Construction of the Embedding Space
>
> This is indeed an important aspect of NOBLE design. The general idea is to form a low-dimensional embedding space that captures biologically relevant neuronal features (e.g., spikecount, AHP width). Training NOBLE with such features enables ensemble response generation by sampling the embedding space during voltage prediction.
>
> Specifically, we define the embedding space by two electrophysiological features (e-features) derived from each HoF model’s frequency-current curve: the threshold current ($I_{thr}$) and the local slope at threshold ($s_{thr}$). These features are normalised to the range [0.5, 3.5] to account for their differing magnitudes. Ithr values are on the order of $10^{-10}$ and $s_{thr}$ values are on the order of 100. This normalisation ensures that the frequency-modulated sinusoidal embeddings remain within the Nyquist limit while preserving periodicity. The construction of the embedding space is detailed in Section 3.3.
>
> ---
>
> ## Necessity of the Embedding
>
> We agree that clearer quantitative metrics are required to demonstrate the advantage of incorporating feature embeddings. To address this, we conducted an ablation study comparing NOBLE's performance as the embedding dimension varies from 0 to 3.
>
> Features embedded|rel L2|Steady State Voltage|Spikecount|AP1 Width|Mean AP Amplitude
> -|-|:-:|:-:|:-:|:-:
> None|12.1%|1.31% |Never fires|Never fires|Never fires
> $s_{thr}$|2.83% |1.33%|4.9%|233%|13%
> $I_{thr}$|2.73%|1.20% |4.4%|107%|14%
> $s_{thr}$, $I_{thr}$|1.92%|1.02%|3.1%|27%|8.9%
> $s_{thr}$, $I_{thr}$, AHP depth|2.16%|1.04%|3.3%|22%|9.5%
>
> These results show that without the use of embeddings, NOBLE is unable to predict any firing response successfully. Embedding either slope or intercept yields similar L2 accuracy, with intercept better capturing AP1 width. Best performance is achieved by embedding both, with minor gains from adding AHP depth. We agree this ablation adds value and included it in the paper
>
> ---
>
> ## Setting Considered
>
> 1. **Low firing rates**. It is indeed more challenging to learn neural activity across a variety of firing regimes. This is actually the setting we consider. We consider neural activity across a realistic range of current inputs from −0.11 to 0.28 nA, which elicits both non-spiking and spiking behaviors (Fig 4). Specifically, Fig 5B demonstrates NOBLE's ability to accurately capture spike frequencies from 0 Hz up to 60 Hz. While our figures primarily highlight the non-spiking and high-firing extremes, we acknowledge that the transition region near the current threshold, with its lower firing rates and higher variability, presents a greater challenge, as seen in the frequency-current curves in Fig 5B. To further address this, we added an explicit example demonstrating NOBLE's robust performance in this challenging, low-spike count regime. We also emphasize that the same model captures the neuronal responses in the different firing regimes.
>
>
> 2. **Time-varying current injections**.  We agree with the reviewer that richer time-varying stimuli would allow for broader validation. We focused on DC step currents because they are widely used in electrophysiological experiments and represent a common protocol for characterising neuron behaviour. More importantly, they offer a readily available source of experimental data to validate NOBLE’s predictions. That said, extending NOBLE to handle arbitrary time-varying inputs is a natural next step that we intend to explore in future work, and added a comment about this at the end of the paper.
>
>
> 3. **Excitatory Neurons**. We appreciate the great observation regarding our focus on inhibitory neurons. Although the NOBLE framework can be applied to both excitatory and inhibitory neurons, we concentrated on inhibitory neurons here due to their significant diversity in morphology, gene expression, and electrophysiology within the human cortex. This diversity presents a particularly rich testbed for evaluating the generalization capabilities of our approach. We already have biophysical PDE-based models for both excitatory and inhibitory neurons, and plan to extend NOBLE to broader populations of neuron types in future work, possibly including excitatory neurons.
>
> ---
>
> ## Clarity of the Exposition
> We appreciate the reviewer’s suggestions for improving the clarity and accessibility of our paper.
>
> ### Limitations section
>
> We thank the reviewer for the suggestion. In response, we added a dedicated Limitations section to the manuscript. This addition improves the clarity of the paper and provides a more complete view of the current constraints of the study. We summarise the main limitations below:
> 1. **Training budget**: While NOBLE enables efficient inference post-training, generating the training dataset requires significant resources to generate the HoF models. We added an ablation study to the paper which shows that NOBLE still performs well when trained on a dataset generated from a smaller number of HoF models
> 2. **Single-cell scope**:  We focus on modelling somatic voltage responses from individual neurons. However, the framework could be extended to multiple neurons by increasing the number of embedded features to capture inter-neuronal variability
> 3. **Setting considered**: As discussed earlier, the current study does not consider time-varying current injections and excitatory neurons. We emphasize once again that this is not a limitation of NOBLE as it extends naturally to these settings, but rather a more general setting we intend to pursue in future work
>
> ### Acronyms
>
> As suggested, we clarified the text by removing certain acronyms. Specifically, we removed FHN (FitzHugh-Nagumo), HH (Hodgkin-Huxley), FFT (Fast Fourier Transform), PINN (physics-informed neural network), and NeRF (Neural Radiance Field).
>
> ### Figures and Captions
>
> We appreciate the feedback and updated the captions, to enhance clarity. Please see the revised versions below.
>
> **Caption 3** Latent representations of neuron models in the normalised ($I_{thr},s_{thr}$)-space. Black dots indicate the 50 training HoF models ({$\mathrm{HoF}^{train}$}), and red crosses represent the 10 test HoF models ({$\mathrm{HoF}^{test}$}) that were excluded during training.
>
> **Caption 4**  NOBLE predictions for current injections of 0.1 nA (top row) and −0.11 nA (bottom row). A) Predicted somatic voltage response for a HoF model in $\mathrm{HoF}^{train}$ previously seen during training, representing an in-distribution case. B) Predicted somatic voltage response for a HoF model in $\mathrm{HoF}^{test}$ not experienced during training, representing an out-of-distribution case.
>
> **Caption 6** Distributions of somatic voltage traces across HoF models for current injections of 0.1 nA (top row) and −0.11 nA (bottom row). From left to right: (1) Ground truth voltage responses from the numerical solver for the 50 training HoF models, $\mathrm{HoF}^{train}$, experienced in training. (2) NOBLE predictions on the same HoFtrain models. (3) NOBLE predictions for 200 randomly sampled models from the convex hull $CH^{train}$.
>
>
> ### Line 96
>
> To clarify Line 96 which describes NOBLE as a flexible framework capable of incorporating a range of ‘interpretable neuron characteristics’, we revised the paragraph beginning at Line 99 to explicitly state that, in the experiments presented here, the latent space is defined by two key e-features: the current at the spiking threshold and its slope
>
> ---
>
> ## Utility of the Fourier Neural Operator (FNO)
>
> The FNO provides a principled and efficient framework for modeling neuronal dynamics by learning mappings from input currents to voltage responses across a broad family of neuron models and current injections. Unlike conventional neural networks that operate on fixed-size vector inputs and outputs, the FNO learns operators, meaning mappings between functions. By operating in the frequency domain, the FNO efficiently captures global, nonlinear, and high-frequency components of voltage responses. Importantly, it generalizes across varying temporal resolutions, different input currents, and different neuron models, enabling accurate simulation of unseen configurations without retraining. This combination of scalability, generalization, and temporal fidelity are typically beyond the reach of standard neural networks. We added a short paragraph at the end of Section 2.2 to emphasize these advantages of the FNO.

---

> > ### Comment · Reviewer_5ihX · 2025-08-03
> > **further question**
> >
> > I thank the authors for addressing thoroughly my questions and comments.
> >
> > I have one further question. Does the selection of particular neural features that are used for embedding dictate which loss function is best for optimising the model? In other words, if authors were to choose another set of neural features, could another type of loss function be better suited to fit the model?

---

> > > ### Author Response · Authors · 2025-08-04
> > >
> > > We thank the reviewer for their positive feedback and for this insightful follow-up question. It highlights an important aspect of our framework’s flexibility, and also points to an interesting direction we are currently working to expand upon in the paper.
> > >
> > >
> > > **Choice of L4 Loss in our paper**
> > >
> > > The choice of the L4​ loss was not dictated by the embedded electrophysiological features (e-features), nor is it central to our framework. The use of L4 for training was motivated by an overfitting study conducted before training on the full dataset where we trained the FNO on a small number of samples using different loss functions to assess their ability to fit voltage traces and preserve key e-features. L4 consistently preserved spike-related features more effectively than L2, particularly in capturing spike amplitudes and widths. Based on these results, we selected L4 as the training loss, but this choice was very specific to the task we conducted.
> > >
> > > In other contexts, alternative loss functions may outperform L4 loss by better capturing features of greater relevance. The choice of loss function should align closely with the specific optimization goals and the aspects of the data most critical for the model to learn.
> > >
> > >
> > > ---
> > >
> > > We would like to expand on that discussion by highlighting an interesting special case. Suppose the goal is to capture the overall dynamics, but with a particular emphasis on accurately modeling one specific neural feature. In such cases, designing the loss function to prioritize this critical metric could lead to improved performance.
> > >
> > > To further illustrate this point, we have actually been working on an additional experiment focused on a special case in which accurately capturing the Action Potential Width (APW) is the primary objective, with greater importance than other features. To address this, we propose two main strategies
> > >
> > > **1. Neuro-inspired Loss**
> > >
> > > For training a model from scratch with a focus on APW, one could incorporate a neuro-inspired term into the loss function, akin to Physics-Informed Neural Operators (PINOs). The total loss would be a weighted sum of the voltage trace error and the feature-specific error,
> > >
> > > $\mathcal{L}=||V-\hat{V}||_4 + \lambda ||APW - \widehat{APW}||_2$.
> > >
> > > This would train the model to simultaneously minimize the error on both the full voltage trace $\hat{V}$ and the predicted feature $\widehat{APW}$. Adjusting the loss coefficient $\lambda$ allows us to control the emphasis placed on accurately capturing the APW feature.
> > >
> > >
> > > **2. Targeted Fine-Tuning**
> > >
> > > On could start from an already pre-trained model, and fine-tune it for a specific feature using only the feature-specific loss,
> > >
> > > $\mathcal{L}=||APW - \widehat{APW}||_2$.
> > >
> > > This approach efficiently adapts the model for a specialized task without requiring retraining from scratch.
> > >
> > > ---
> > >
> > > More advanced multi-objective optimization techniques have also been developed for PINNs and PINOs, and can be leveraged in this context to enhance these strategies.
> > >
> > > To validate these approaches, we are currently performing a fine-tuning experiment targeting the APW feature, and should have results in the next few weeks. As these results will not be available within the next few days, we regret that we cannot comment on their effectiveness during the discussion period. However, we will carefully analyze the findings and incorporate them into the paper as soon as they are ready.
> > >
> > >
> > > ---
> > >
> > > We hope this clarifies our methodology and the flexibility of the proposed framework. Is there anything else from our previous response that we can clarify further, or any additional questions or concerns that we can address?

---

### Official Review · Reviewer_tkUK · 2025-06-30

**Clarity:** 3
**Significance:** 4
**Originality:** 3
**Rating:** 5
**Confidence:** 3

**Summary:**

This paper proposes NOBLE, a neural operator framework based on Fourier Neural Operators (FNO), designed to efficiently approximate detailed multi-compartment neuron simulations. By training on neuron models, the method learns to predict voltage traces, achieving substantial speed-ups over traditional PDE solvers while maintaining biophysical interpretability. It also enables interpolation across different parameter regimes, making it well-suited for integration into a variety of neuroscience modeling studies.

**Questions:**

1. Why was the L4 error function chosen as the training objective?
2. In Figure 5A, why are there no black lines for the 'Experiment' data, and what do the black x markers indicate?
3. NOBLE produces noisier outputs compared to the PDE solution (e.g, voltage traces in Figure 6). If training were optimal, should these fluctuations be eliminated? Do they reflect imperfect training?
4. Conducting simple ablation tests that vary training hyperparameters would help assess the robustness of the FNO training.
5. How does NOBLE have trial-to-trial variability? Where does the variability come from (e.g., noise injection or random sampling)?

**Ethical Concerns:**

["NO or VERY MINOR ethics concerns only"]

**Final Justification:**

I appreciate the authors’ detailed responses and the additional experiment provided during the discussion. I am keeping my score unchanged, as my initial evaluation was already positive.

**Limitations:**

Yes

**Paper Formatting Concerns:**

not observed.

**Quality:**

3

**Strengths And Weaknesses:**

**Strengths**
1. **Quality and significance**: This method is technically sound and can be used across a wide range of research.
2. **Clarity**: The manuscript is well written.

**Weaknesses**
1. **Lack of real experiment**: This paper provides experiments based on comparing neuron models, without direct validation with real experimental data.
2. **Narrow set of benchmarks**: Evaluation is limited to comparisons with the PDE solver, without any learning-based models. While interpolation-based comparison may be ambiguous, direct comparison of fitting performance should be possible.
3. **Generalization to other neuron types**: All experiments focus on PVALB neuron models. The out-of-distribution samples (blue circle in Figure 12) remain close to the point that is used for training.

     However, this may be acceptable, as the authors conducted extensive experiments to demonstrate the method’s reliability within the PVALB neuron domain.
.

---

> ### Author Rebuttal · Authors · 2025-07-31
>
> We sincerely thank the reviewer for their thoughtful and constructive feedback. We're especially grateful for the recognition of our method’s technical strength, wide applicability, and clear presentation. We have addressed all raised concerns below and hope our response satisfactorily resolves the reviewer’s points. We are happy to provide further elaboration or modifications during the discussion period
>
> ---
>
> **W1: Validation against real data**
>
> Unlike other existing ML papers [28-33;37-40] in neuroscience focusing exclusively on synthetic neuron dynamics simulations, we validate our model against real in vitro data in Fig 7. We compare NOBLE's predicted e-features to those extracted from real biological human neuron recordings. We report relative errors for both the biophysical models and NOBLE against experimental data for the most important e-features, and show that NOBLE exhibits good performance in predicting key e-features of real experiments when compared to the relative error from fully detailed bio-realistic models
>
> ---
>
> **W2: Benchmarks**
>
> We understand the concern about limited comparisons to baselines. Our primary goal is to evaluate whether the neural operator framework can capture the biological variability seen in real-world experiments. To this end, we used data generated from PDE-based models which were selected via evolutionary multi-objective optimization as the best available approximations of experimental recordings, and exhibit the desired variability. Now, other existing ML approaches [28-33;37-40] in neuroscience are not equipped to capture variability, and can not be used for a comparison. While other models could be used within NOBLE and trained on the same data, we believe this would not offer additional insight
>
> Our focus is on learning representations that reliably capture key e-features across neuron dynamics and parameter regimes, and NOBLE demonstrates strong performance on this. Since the PDE solvers generating the dataset themselves are approximations of biological reality with non-negligible errors, further reducing prediction errors would lead to further overfitting to the PDE solver simulations but probably not to substantially better-captured dynamics and e-features when compared to true real-life recordings. For this reason, we tuned hyperparameters up to the level of error seen between the PDE models and experimental data, as marginal error improvements on the PDE data are unlikely to translate meaningfully when compared to real experimental data. We included a note on this in Section 4.1 of the revised paper
>
> ---
>
> **W3: NOBLE generalizes beyond its training distribution**
>
> Fig 6 (rightmost plot) shows that NOBLE generalizes well across the entire convex hull of HoF models, not just near HoF models experienced during training. This highlights NOBLE's significance as the first approach capable of generating voltage responses for unseen HoF models that remain within the distribution of valid outputs produced by the PDE solver
> To further address concerns about broader applicability, we evaluated NOBLE on a different neuron type: the vasoactive intestinal peptide (VIP) interneuron. We tested two settings:
> * Cross-type generalisation, applying the PVALB-trained NOBLE directly to a VIP neuron. Achieves errors: Relative L2: 14%, Spike count: 8.0%, AP1 width: 62%, Mean AP amplitude: 29%
> * Training a new NOBLE model on VIP neurons. Achieves errors: Relative L2: 2.5%, Spike count: 9.2%, AP1 width: 23%, Mean AP amplitude: 10%
>
> These results show that NOBLE generalizes modestly across cell types but performs best when trained within a specific type. Its strong performance on both PVALB and VIP neurons highlights the generalizability of the embedding framework. Full results are provided in the updated appendix
>
> ---
>
> **Q1: L4 loss was chosen based on empirical performance in early experiments**
>
> Before training on the full dataset, we conducted (added to the appendix) an ablation study to assess the effectiveness of various loss functions in capturing voltage traces and e-features. We performed a grid search by training the model on a small subset of samples, providing an estimate of each loss function’s best possible performance under ideal conditions. L4 consistently preserved spike-related features more effectively than L2, so we selected L4 as the training loss
>
> ---
>
> **Q2: Figure 5A**
>
> The black ‘x’ markers represent experimental firing rates measured at discrete current amplitudes. Unlike the PDE simulations, which cover a finely spaced input range, experimental recordings were limited to a small number of amplitudes due to the temporal constraints of in vitro electrophysiology (one can only keep a neural recording stable during an in vitro experiment for ~20 minutes, see e.g. [13], limiting the number of experiments that can be recorded with repetition). We chose not to connect these points with a line, as the data is too sparse, but clarified this choice in the caption
>
> ---
>
> **Q3: Fluctuations in NOBLE’s predictions**
>
> We first point out that the oscillations are low in amplitude and do not significantly affect the extracted e-features. Moreover, similar fluctuations can be observed in real experimental voltage recordings, supporting their biological plausibility
>
> Now, capturing high-frequency spiking activity requires the FNO to leverage its higher frequency modes. In contrast, non-firing regimes involve slower dynamics, where these high-frequency components are unnecessary. In attempting to reconcile both regimes within a single model, the FNO relies on interference between higher frequency modes to cancel out their contributions during non-firing activity. However, this cancellation is imperfect, resulting in small residual oscillations. Training a model specifically for non-firing responses would reduce or eliminate these artifacts, but this comes at the cost of generalization. We chose to prioritize a unified model capable of handling the full range of neuronal behaviors, accepting minor imperfections in favor of broader applicability.
>
> ---
>
> **Q4: Ablation Studies**
>
> We agree with the reviewer’s view on the importance of including ablation studies. As discussed in our response to W2, we tuned hyperparameters up to the level of error seen between PDE models and experimental data. At this level, marginal error differences on the PDE data are unlikely to translate meaningfully when compared to real experimental data, limiting the significance of ablation studies on certain model and training hyperparameters. However, we added to the revised version two additional ablation studies on key components of NOBLE.
>
> **1. Feature embedding ablation**
>
> To evaluate the benefits of using e-feature embeddings, we vary the embedded features used in 5 different ways:
>
> * No embedded features
> * $I_{thr}$
> * $s_{thr}$
> * $s_{thr}$, $I_{thr}$
> * $s_{thr}$, $I_{thr}$, AHP depth
>
> Features embedded|rel L2|Steady State Voltage|Spikecount|AP1 Width|Mean AP Amplitude
> -|-|:-:|:-:|:-:|:-:
> None|12.1%|1.31% |Never fires|Never fires|Never fires
> $s_{thr}$|2.83% |1.33%|4.9%|233%|13%
> $I_{thr}$|2.73%|1.20% |4.4%|107%|14%
> $s_{thr}$, $I_{thr}$|1.92%|1.02%|3.1%|27%|8.9%
> $s_{thr}$, $I_{thr}$, AHP depth|2.16%|1.04%|3.3%|22%|9.5%
>
> These results show that without the use of embeddings, NOBLE is unable to predict any firing response successfully. Embedding either slope or intercept yields similar L2 accuracy, with intercept better capturing AP1 width. Best performance is achieved by embedding both, with minor gains from adding AHP depth. We agree this ablation adds value and included it in the paper
>
> **2. HoF Population Ablation**
>
> We vary the number of HoF models used to generate the training data to investigate the importance of variability and diversity of dynamics. Specifically, we constructed training sets using different numbers of HoF models, ensuring that the total number of samples used in training remained the same. This enables an evaluation of the initial cost associated with ensuring a sufficient number of available HoF models
>
> #HoFs excluded|rel L2|Steady State Voltage|Spikecount|AP1 Width|Mean AP Amplitude
> -|-|:-:|:-:|:-:|:-:
> 10|11.7%|2.0%|9.2%|350%|14%
> 20|10.9%|1.8%|19%|920%|17%
> 30|10.9%|1.9%|20%|3004%|20%
> 40|10.6%|1.9%|45%|1698%|20%
>
> Although the relative L2 error on traces remains stable, there is a degradation in spike-related features, particularly spike count and AP1 width, as training diversity decreases. This ablation has been added to the paper as it highlights the importance of model diversity, not just quantity, for learning robust representations of neural dynamics
>
> ---
>
> **Q5: Trial-to-trial variability arises from the diversity across HoF models**
>
> We thank the reviewer for raising this important question.The trial-to-trial variability arises from the structured diversity among the biophysical models used for training. The HoF models are generated using an evolutionary optimization procedure that searches for PDE parameters whose simulations, through BMTK, closely match experimental recordings. Since the experimental data naturally exhibit trial-to-trial variability and the HoF models are obtained using random subsets of the entire collection of experimental recordings, the resulting population of optimized PDEs captures a range of biologically plausible dynamics. Simulating across this population reproduces the variability observed in repeated biological trials.
>
> NOBLE can emulate the HoF models, and thus reproduce the structured variability present in the training distribution. We emphasize that our embedding is the key to capture variability, and that prior ML approaches [28-33;37-40] in neuroscience are not equipped to capture variability. At inference time, conditioning NOBLE on different HoF embeddings results in output variations that reflect different realistic biological responses. We updated Section 3.3 to clarify how the trial-to-trial variability is achieved within NOBLE

---

> > ### Comment · Reviewer_tkUK · 2025-08-05
> >
> > I appreciate the authors’ detailed responses and the additional experiment provided during the discussion. My concerns and questions were well addressed, and I believe the clarifications in the response will help strengthen the manuscript. I am keeping my score unchanged, as my initial evaluation was already positive.

---

### Decision · Program_Chairs · 2025-09-17

**Decision:**

Accept (poster)

**Comment:**

This paper presents NOBLE, a framework for predicting single neuron voltage responses to injected currents. The method is based on an artificial neural network that learns to approximate the solutions the results of simulations of multi-compartmental models. NOBLE defines a latent space of interpretable neuronal features that enables meaningful interpolation between different parameter regimes and efficient generation of synthetic neurons with realistic behavior.

The reviewers appreciated the relevance of the work to the NeurIPS community and its potential impact for a broad range of computational neuroscience applications, as well as its technical soundness. The main concerns raised by the initial reviews included a lack of more extensive benchmarking against other methods and types of data, along several requests for clarification about the details of the method and its relationship with other approaches. The authors responded by providing additional explanations and analyses, testing their method on an additional neuronal type (as the original manuscript only included one type) and arguing why comparisons with other learning-based methods may not be fair/provide additional insight. A substantial discussion with one of the reviewers focused on the merits and novelty of the proposed method vs a specific prior work (NeuPRINT).

Overall, following also the internal discussion among reviewers, the consensus is that the merits of the paper and its potential usefulness to the community outweigh its limitations. I recommend acceptance.